# WHEN TO ENSEMBLE: IDENTIFYING TOKEN-LEVEL POINTS FOR STABLE AND FAST LLM ENSEMBLING

**Heecheol Yun[1], Kwangmin Ki[2], Junghyun Lee[1], Eunho Yang[1,3]**[*]
[1]KAIST, [2]Korea University, [3]AITRICS
{yoon6503, eunhoy}@kaist.ac.kr

## ABSTRACT

Ensembling Large Language Models (LLMs) has gained attention as a promising approach to surpass the performance of individual models by leveraging their complementary strengths. In particular, aggregating models' next-token probability distributions to select the next token has been shown to be effective in various tasks. However, while successful for short-form answers, its application to long-form generation remains underexplored. In this paper, we show that using existing ensemble methods in long-form generation requires a careful choice of ensembling positions, since the standard practice of ensembling at every token often degrades performance. We identify two key factors for determining the ensembling positions: tokenization mismatch across models and consensus in their next-token probability distributions. Based on this, we propose **SAFE**, (**S**table **A**nd **F**ast LLM **E**nsembling), a framework that selectively ensembles by jointly considering these factors. To further improve stability, we apply a probability sharpening strategy when the ensemble distribution becomes overly smooth, enabling the selection of more confident tokens during ensembling. Our experiments on diverse benchmarks, including MATH500 and BBH, demonstrate that SAFE outperforms existing methods in both accuracy and efficiency, with gains achieved even when ensembling fewer than 1% of tokens. The code is available at https://github.com/yoon6503/SAFE.

## 1 INTRODUCTION

Recently, Large Language Models (LLMs) have achieved remarkable performance across diverse domains, including mathematics (Yang et al., 2024b), coding (Guo et al., 2024) and reasoning (Yang et al., 2025; OpenAI, 2024). Despite this progress, each LLM possesses unique strengths shaped by its training recipe, and no single model dominates across all domains. As a result, combining the complementary strengths of multiple models at inference time has emerged as a promising way to surpass the performance of any individual model (Wang et al., 2025a; Yao et al., 2025; Chen et al., 2025a). Compared to training a new model that jointly integrates all such capabilities, these collaborative approaches provide a more practical and efficient pathway to superior performance.

Among various training-free collaboration methods, *probability-level ensemble*, which aggregates the next-token probability distributions of multiple LLMs to select the most confident token, has emerged as one of the most effective ways (Yao et al., 2025; Yu et al., 2024; Huang et al., 2024; Xu et al., 2024). It enables collaboration across diverse model architectures and effectively leverages the knowledge of multiple models embedded in their probability distributions. Consequently, it has outperformed individual models, particularly when directly answering multiple-choice or short-answer questions without reasoning.

A natural question then arises: *are probability-level ensemble methods equally effective for long-form generation?* We find that, in long-form generation, the effectiveness of ensembling critically depends on deciding *when* to ensemble. Our analysis reveals that accuracy and efficiency improve when ensembling occurs at appropriate token positions, guided by two key factors: **tokenization mismatch** across models and their **consensus in next-token probability distributions**.

---

[*]Corresponding author.

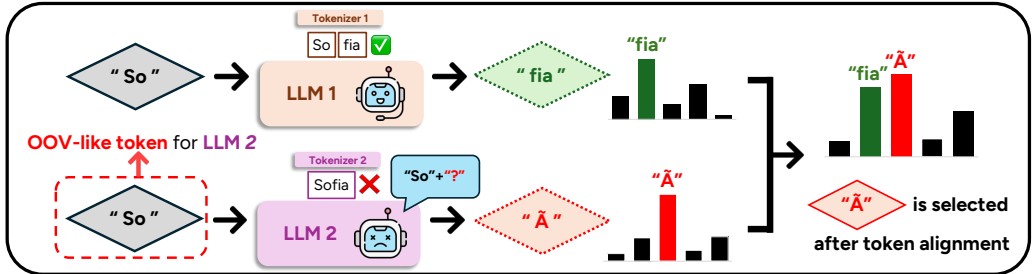

Figure 1: Illustration of the OOV-like token problem. When `So` is fed into $LLM_2$, which tokenizes `Sofia` as a single token, the probability distribution of the next token becomes corrupted.

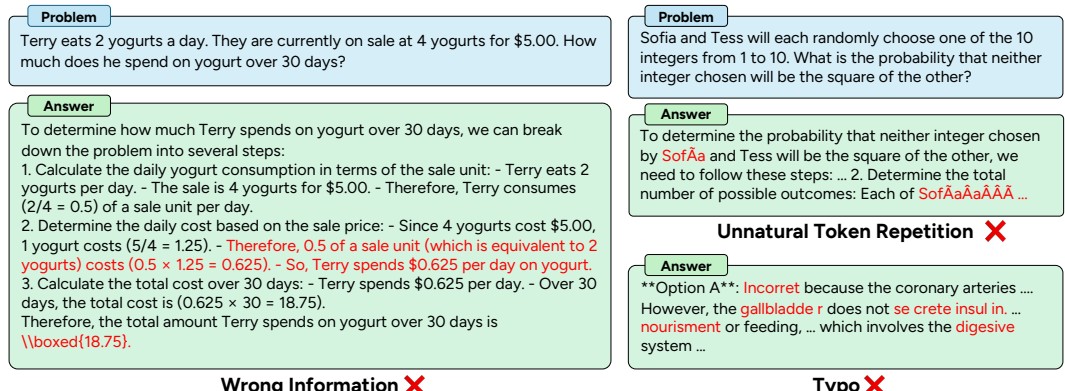

Figure 2: (Left) Failure cases of existing ensemble methods in long sequence generation. (Right) Feeding OOV-like tokens into a model often causes it to produce wrong tokens.

The first factor, **tokenization mismatch across models**, is crucial for *stability*, especially in long-form generation where such mismatches occur more frequently. A mismatch arises when an ensemble selects a token that conflicts with the tokenization scheme of a participating model. We refer to these tokens as OOV-like tokens because while not truly out-of-vocabulary (OOV), they force a model to predict the next token in an out-of-distribution state, often resulting in the generation of erroneous tokens, as illustrated in Figure 1. Suppose the ensemble process first generates the token `So` when constructing the word `Sofia`. However, since $LLM_2$ tokenizes `Sofia` as a single token, `So` acts as an OOV-like token for $LLM_2$. Conditioning on this unnatural prefix corrupts $LLM_2$'s next-token probability distribution, leading to an erroneous output (*i.e.*, Ã). Such errors accumulate in long-sequence generation, degrading output quality. For example, in the case of unnatural token repetition shown in Figure 2, an initial error in generating the word `Sofia` propagates, causing the model to repeatedly output corrupted tokens like "Ã" on the subsequent generation. Consequently, existing ensemble method (Yao et al., 2025) that performs ensembling at every token, suffers substantial performance degradation in Chain-of-Thought (CoT) reasoning (Wei et al., 2022), as shown in Table 1. Therefore, accounting for tokenization mismatch across models is essential to prevent the introduction of OOV-like tokens and ensure stable ensembling under CoT.

The second factor, **consensus in models' next-token probability distributions**, relates to *efficiency*. Given the next-token probability distributions from multiple models, an ensemble operation is performed to aggregate these probability distributions. However, this introduces inefficiency when generating long sequences because the number of ensemble operations grows with sequence length. The primary expense of the ensemble operation arises from aligning next-token probability distributions defined over different vocabularies into a shared vocabulary space, a process that requires mapping across large vocabulary sets. However, when individual models' next-token probability distributions exhibit sufficient consensus, the most confident token from the aggregated distribution can be identified without explicitly aligning distributions from multiple models. Leveraging this property, we can determine the most confident token directly from the models' next-token probability distributions, thereby improving efficiency by skipping alignment operations.

| Method | MMLU-redux | | ARC-C | | MATH500 |
|---|---|---|---|---|---|
| | **No CoT** | **CoT** | **No CoT** | **CoT** | **CoT** |
| Qwen2.5-7B | 68.86 | 74.88 | 87.37 | 88.74 | 72.4 |
| Internlm3-8B | 67.52 | 76.89 | 88.57 | 90.27 | 74.8 |
| UniTE | 69.36 (+0.50) | 73.39 (-3.50) | 88.40 (-0.17) | 87.97 (-2.30) | 59.6 (-15.2) |
| **UniTE + Ours** | 69.36 (+0.50) | **77.92** (+1.03) | 88.40 (-0.17) | **90.78** (+0.51) | **77.6** (+2.8) |

Table 1: Performance of the baseline ensemble method (UniTE) degrades under CoT prompting. In contrast, it matches or outperforms individual models when directly answering multiple-choice questions, since tokenizer mismatches do not arise. All models are instruction-tuned.

To this end, we propose **SAFE** (**S**table **A**nd **F**ast LLM **E**nsembling), which identifies the opportune moments for ensembling in long-sequence generation by considering the two key factors above. SAFE adopts a speculative strategy in which one model, the *drafter*, generates a lookahead sequence of tokens, while the remaining models, the *verifiers*, identify token-level ensemble points within that sequence. Similar to speculative decoding (Leviathan et al., 2023), this role separation reduces computational cost by limiting autoregressive generation to the drafter, whereas conventional ensemble methods require every model to do so. Specifically, SAFE iterates a three-step cycle: Generate–Verify–Ensemble. (**Generate**) First, the drafter produces a lookahead sequence of tokens. (**Verify**) Next, the verifiers examine drafter's tokens in a single forward pass to determine whether ensembling at each token is both stable and necessary. Ensembling is triggered among the drafter's tokens only when the following two conditions are satisfied: (i) OOV-like token is not introduced and (ii) the verifiers exhibit insufficient agreement on the token. (**Ensemble**) Finally, ensembling is applied only at the tokens validated in the **Verify** step, replacing them with the ensembled tokens. At these points, if the ensemble distribution is overly smooth, we apply a probability sharpening strategy that concentrates the probability mass onto the most plausible token for precise token selection.

Overall, we find that probability-level ensembling should occur at appropriate token positions, especially when generating long sequences with models that use different tokenizers. We then propose SAFE, a method that determines these positions by jointly considering the two key factors. Consequently, our method offers the following key advantages.

- **Efficiency**: SAFE significantly reduces computational cost in two ways. First, its speculative strategy restricts costly autoregressive generation to a single drafter. Second, its selective ensembling reduces the number of ensemble operations. Therefore, SAFE can achieve inference speed comparable to individual models, even on long sequences.

- **Stability**: SAFE ensures that tokens are generated from an uncorrupted ensemble distribution by preventing OOV-like tokens from being fed into models. As a result, SAFE enables stable text generation and outperforms existing ensemble methods in CoT settings.

- **Plug-and-Play**: SAFE can be seamlessly integrated with existing ensemble methods by simply adding the generate-verify logic. SAFE consistently improves recent ensembling approaches across diverse model combinations.

## 2 RELATED WORK

### 2.1 LLM ENSEMBLE

LLM ensemble methods can be broadly categorized according to whether ensembling occurs **after inference** or **during inference** (Chen et al., 2025b). Research in both directions has progressed in parallel, each of which is detailed below.

#### 2.1.1 ENSEMBLE AFTER INFERENCE

These approaches aggregate complete responses from multiple LLMs. While early work relied on debate-style interactions to reach consensus (Du et al., 2023; Chen et al., 2024a; Liang et al., 2024), recent focus has shifted toward stacking LLMs in cascade, parallel, or hybrid structures.

**Cascade structure.** Models are invoked sequentially to balance cost and performance. Frugal-GPT (Chen et al., 2024b) orders models by cost, invoking more expensive models only when cheaper ones produce unreliable responses. Similarly, AutoMix (Aggarwal et al., 2024) uses self-verification to guide routing decisions, while Gupta et al. (2024) employs token-level uncertainty.

**Parallel structure.** In contrast to cascading, parallel ensembling runs multiple models independently and then selects the best response among them. MORE (Si et al., 2023) trains a classifier to select the optimal response by considering model expertise, confidence, and agreement across responses. LLM-Blender (Jiang et al., 2023) employs a pairwise ranker to score responses and then fuses the top-$k$ candidates into a single answer.

**Hybrid structure.** Recent work has also explored combining the advantages of cascade and parallel structures. MoA (Wang et al., 2025a) proposes a framework that iteratively feeds responses from multiple models into an aggregator LLM, which consolidates them into a single answer. Within this framework, Self-MoA (Li et al., 2025) shows that using a single best-performing model can outperform using multiple distinct models in some cases, while Symbolic-MoE (Chen et al., 2025a) introduces query-adaptive model routing. Nevertheless, such frameworks require numerous LLM calls, and consolidating multiple responses remains challenging (Wang et al., 2025b).

### 2.1.2 ENSEMBLE DURING INFERENCE

In this setting, ensembling occurs during response generation, most commonly at the token level. Co-LLM (Shen et al., 2024) adopts a routing method which dynamically selects which model to use for generating each token. CoSD (Wang et al., 2025c) improves efficiency by introducing a lightweight router and integrating speculative decoding. These approaches primarily target models with identical tokenizers and rely on routing rather than aggregating probability across models.

To better exploit the collective intelligence of multiple models, another line of work explores *probability-level ensemble* methods. These methods average the next-token probability distributions of different models to select the most confident token. Since probability distributions are defined over heterogeneous vocabularies, prior work has focused on constructing the ensemble distribution by aligning different vocabularies across models. GaC (Yu et al., 2024) integrates probabilities by taking the union of all model vocabularies and then mapping each model's vocabulary to this union. DEEPEN (Huang et al., 2024) projects each model's vocabulary into a shared embedding space, merges distributions there, and maps them back to the individual vocabulary spaces. UniTE (Yao et al., 2025) demonstrates that aligning only the top-$k$ tokens from each model is effective both in performance and efficiency. While these methods achieve strong performance in directly generating answer tokens by selecting the most confident token, they face challenges in long-sequence generation that involves reasoning. In such cases, an increase in OOV-like tokens destabilizes the ensemble, and repeated autoregressive generation across multiple models, along with the need to align their vocabulary spaces, makes such approaches inefficient. Therefore, we aim to simultaneously improve the stability and efficiency of probability-level ensembling by introducing a verification algorithm that determines *when* to ensemble.

### 2.2 SPECULATIVE DECODING

Speculative decoding (Leviathan et al., 2023) is a widely used technique for reducing the cost of autoregressive generation in LLMs. To alleviate the repeated forward passes required by a large model during token generation, speculative decoding replaces this process with a small *drafter* that speculates a sequence of candidate tokens. The large target model then performs a single forward pass to determine how many of the drafter's proposed tokens to accept. This allows multiple tokens to be generated in a single forward pass of the target model, thereby reducing computational cost.

Recently, speculative decoding has been explored as a way to accelerate probability-level LLM ensembling. However, existing approaches (Fu et al., 2025) are limited to settings in which all models share an identical tokenizer, and cases where the drafter and target models use different tokenizers remain underexplored. In such scenarios, the drafter's tokens cannot be properly evaluated by other participating models due to tokenization misalignment and also exhibit OOV-like issues. To address this, we extend speculative decoding to ensembles composed of models with heterogeneous tokenizers by proposing an appropriate acceptance criterion.

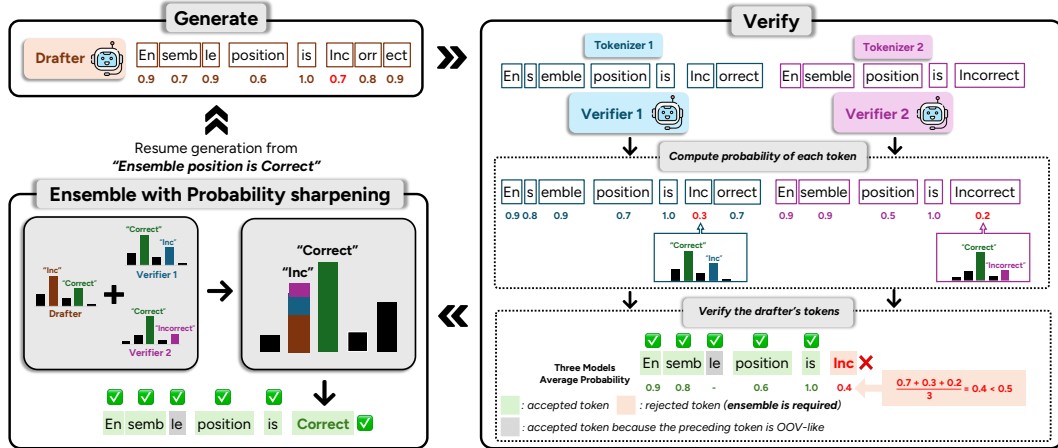

Figure 3: Overview of SAFE. The drafter generates a sequence of tokens, which the verifiers re-tokenize with own tokenization schemes and identify the necessary ensembling point. At this position, ensembling is performed with probability sharpening applied selectively to enhance precision.

## 3  SAFE: Toward Stable and Fast LLM Ensembling

We aim to solve the problem of instability and inefficiency that arises when ensembling next-token probability distributions across LLMs with heterogeneous tokenizers, particularly in long-sequence generation. To this end, we propose **SAFE**, an algorithm that preemptively determines optimal points for ensembling by jointly considering tokenization mismatch and consensus in next-token probability distributions. Notably, SAFE can be seamlessly integrated with existing ensemble methods.

Given $k$ different LLMs, our method begins by dividing the models into two roles: a drafter $M_{\text{draft}}$, which generates a lookahead sequence of tokens, and verifiers $M_{\text{ver}}$, which identify the ensemble points among the drafter's tokens. We select the best-performing model as $M_{\text{draft}}$, while the remaining models serve as $M_{\text{ver}}$. We then iterate the **Generate** (Section 3.1)-**Verify** (Section 3.2)-**Ensemble** (Section 3.3) cycle. In each iteration, $M_{\text{draft}}$ first generates a sequence of tokens, which $M_{\text{ver}}$ then examine to find a token that requires ensembling. At such points, ensembling is performed to replace the token with the most confident token from the averaged distributions of all models, after which $M_{\text{draft}}$ resumes generation from the ensembled token. Figure 3 shows the overview of SAFE.

### 3.1  Generate

The drafter $M_{\text{draft}}$ generates a predefined number $n$ of tokens $(t_i, \dots, t_{i+n-1})$. Producing multiple tokens rather than a single token allows SAFE to account for the different tokenization schemes of the various models. For example, consider the word `Incorrect`. Suppose $M_{\text{draft}}$ generates it as three tokens (`Inc`, `orr`, `ect`), while other models generate it as a single token (`Incorrect`). If $M_{\text{draft}}$ were to generate only the first token `Inc`, it would fail to capture the tokenization schemes of the other models. Therefore, at this stage, $M_{\text{draft}}$ produces a sequence of tokens to ensure compatibility with diverse tokenization schemes. The choice of $n$ is discussed in Section 4.4.

### 3.2  Verify

In this step, the verifiers $M_{\text{ver}}$ collaboratively examine the drafter's tokens $(t_i, \dots, t_{i+n-1})$ to identify which tokens require ensembling. Ensembling is triggered at the earliest token $t_j$ that satisfies two checks: (i) *OOV-like token verification*, requiring that the immediately preceding token $t_{j-1}$ is not an OOV-like token, and (ii) *ensemble distribution verification*, requiring that $t_j$ is not the most confident token in the ensemble distribution $P_{\text{ens}}$. This selective process addresses instability by preventing the introduction of OOV-like tokens and improves efficiency by skipping unnecessary ensembling. Importantly, this entire verification process is efficient, as the drafter's tokens are processed by the verifiers in a single forward pass rather than autoregressively.

**(i) OOV-like Token Verification.** This check requires that the preceding token is not an OOV-like token to prevent OOV-like tokens from corrupting the model's next-token probability distribution. To determine whether a drafter token $t_j$ is an OOV-like token, we examine whether the token boundary up to $t_j$ aligns with the tokenization boundaries of the verifiers, ensuring that each verifier can be conditioned on valid prefix tokens. For example, in the word `Incorrect`, tokens such as `Inc` or `orr` are OOV-like, but `ect` is not. This is because the tokenization boundaries up to `Inc` or `orr` are inconsistent with the other model's tokenization boundary `Incorrect`, forcing that model to be conditioned on an invalid prefix such as `Inc` or `Incorr`. The detailed verification process are as follows. First, each verifier model $LLM_v \in M_{\text{ver}}$ tokenizes the drafter's sequence $\mathbf{t}_{<i+n}$ into its own tokenization, $\mathbf{t^v}_{<v_{i+n}}$. Then, the drafter's token $t_j$ is defined as an OOV-like token for $LLM_v$ if the tokenization boundary of $\mathbf{t}_{<j+1}$ does not match any boundary in $LLM_v$'s tokenization. This condition is formally stated in Equation (1):

$$t_j \text{ is OOV-like in } LLM_v \iff \forall x \in [0, v_{i+n}-1], \quad \text{Decode}(\mathbf{t}_{<j+1}) \neq \text{Decode}(\mathbf{t^v}_{<x+1}), \quad (1)$$

where $\text{Decode}(\cdot)$ means merging tokens back into text. If $t_j$ is identified as OOV-like by any verifier, ensembling is not triggered at the subsequent token $t_{j+1}$. Therefore, in the word `Incorrect`, ensembling is skipped at `orr` and `ect`, but can be triggered at the token following `ect`.

**(ii) Ensemble Distribution Verification.** For tokens that pass OOV-like token verification, our method further checks whether the token is the most probable prediction in the ensemble distribution. To avoid the cost of repeatedly constructing the ensemble distribution, we instead verify whether a token $t_j$ is the most probable token by examining each model's own distribution. Specifically, given $LLM_v$'s tokenization $\mathbf{t^v}_{<v_j}$ of the drafter's tokens $\mathbf{t}_{<j}$, the drafter's token $t_j$ is regarded as the most confident and ensembling is therefore skipped, if either of the following holds:

1. **(Unanimous consensus among verifiers)**
   If $t_{v_j}^v = \arg\max_t P_v(t \mid \mathbf{t^v}_{<v_j})$ for all $LLM_v \in M_{\text{ver}}$, we skip ensembling, where $P_v$ is the probability distribution of $LLM_v$.

2. **(Average probability above one half)**
   If $\frac{1}{|M_{\text{ver}} \cup M_{\text{draft}}|} \sum_{LLM_v \in M_{\text{ver}} \cup M_{\text{draft}}} P_v(t_{v_j}^v \mid \mathbf{t^v}_{<v_j}) > \frac{1}{2}$, we skip ensembling at $t_j$.

Intuitively, the first condition checks whether $t_{v_j}^v$ is the most probable one across all verifiers, while the second checks whether its average probability across all models is greater than 0.5. Adopting these criteria does not compromise accuracy compared to using the exact ensemble distribution, which is proved in Appendix C.

---

**Algorithm 1 SAFE**: Generate-Verify-Ensemble algorithm

---

**Require:** $M_{\text{draft}}, M_{\text{ver}}, p$: prompt, $n$: drafter's sequence length
1: $i \leftarrow 1$
2: $t_0 \leftarrow BOS$ token
3: **while** not End-of-Sentence **do**
4:      $t_i, \cdots, t_{i+n-1} \leftarrow M_{\text{draft}}(p, \mathbf{t}_{<i})$          ▷ **1. Generate**
5:      $\mathbf{t^v}_{<v_{i+n}} \leftarrow \text{TOKENIZE}_v(\mathbf{t}_{<i+n}), \quad \forall v \in M_{\text{ver}} \cup M_{\text{draft}}$
6:      **for** $j = i \rightarrow i+n-1$ **do**          ▷ **2. Verify**
7:          **if** $t_{j-1} \neq$ OOV-like token **and** $t_j$ passes Ensemble Distribution Verification **then**
8:              $P_{\text{ens}} \leftarrow \text{AVERAGEDIST}(\{P_v(\cdot \mid p, \mathbf{t^v}_{<v_j})\}_{v \in M_{\text{ver}} \cup M_{\text{draft}}})$      ▷ **3. Ensemble**
9:              $t_j \leftarrow \arg\max_t P_{\text{ens}}(t \mid p, \mathbf{t}_{<j})$
10:              $i \leftarrow j + 1$
11:          **else**
12:              $i \leftarrow i + n$
13:          **end if**
14:      **end for**
15: **end while**

---

## 3.3 ENSEMBLE: SHARPENING ENSEMBLE DISTRIBUTION

In the **Ensemble** step, any token that passes both verifications in the **Verify** step is replaced with the most probable token from $P_{\text{ens}}$, which is constructed as the average of all models' probability distributions using existing ensemble methods. However, different tokenization schemes across

models can scatter probability mass for the same word across multiple sub-word tokens, resulting in an overly smooth ensemble distribution (*i.e.*, $\max P_{\text{ens}} < 0.5$) that hinders confident token selection. To address this, we apply a probability sharpening strategy to consolidate the diffused probability mass. We explore two different sharpening strategies. The first adopts a heuristic approach that consolidates the diffused probability by reallocating the probability mass from variant subword tokens to their common prefix token. To prevent inflating probabilities of low-quality tokens, reallocation is applied only to the drafter's tokens with initial probability greater than a threshold $\lambda$. Formally, the entire sharpening process is defined as:

$$P_{\text{ens}}(t_j) \leftarrow P_{\text{ens}}(t_j) + \sum_{t_i \, : \, t_i.\texttt{startswith}(t_j)} P_{\text{ens}}(t_i), \text{ where } P_{\text{ens}}(t_j) > \lambda.$$

The second strategy replaces the arithmetic mean with the geometric mean when aggregating the models' probability distributions. Since the geometric mean strongly penalizes tokens that receive low probability by any individual model, it effectively concentrates probability mass on tokens that are consistently supported across models. These two strategies are compared in section 4.4.

After selecting the most confident token from the (potentially sharpened) ensemble distribution, the drafter resumes generation from this token.

## 4 EXPERIMENTS

This section evaluates **SAFE** across various benchmarks and model combinations. We begin by outlining the experimental setup, followed by the implementation details, including our KV caching strategy to further improve efficiency. We then provide an analysis of the results.

### 4.1 EXPERIMENTAL SETTINGS

**Models.** We select three widely used LLMs with similar capability but heterogeneous tokenization schemes: Internlm3-8B-Instruct (Cai et al., 2024), Qwen2.5-7B-Instruct (Qwen et al., 2025), and EXAONE-3.5-7.8B-Instruct (An et al., 2024). Figure 4 presents the tokenization similarity across model pairs on Oxford 5000 words (Oxford, 2018), which consists of commonly used English words. As illustrated, only a small portion of the words are tokenized identically across models, with agreement rates ranging from 40% to 60%. To also evaluate ensembling on models with nearly identical tokenizations, we include two widely used LLMs with high agreement rates: Qwen2-7B-Instruct (Yang et al., 2024a) and Llama-3.1-8B-Instruct (Dubey et al., 2024). For these two models, more than 99% of Oxford 5000 words are tokenized identically. For further study, 32B-scale models are experimented in Appendix G.

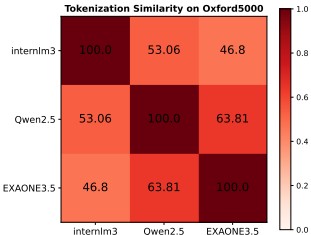

Figure 4: Tokenization agreement rates between each model pair on Oxford 5000 words.

**Benchmarks.** To evaluate performance across diverse domains, we use five benchmarks. For general knowledge, we adopt MMLU-redux (Gema et al., 2025), a refined subset of MMLU (Hendrycks et al., 2021) that covers 30 subjects with human-annotated corrections. For mathematical reasoning, we use MATH500 (Lightman et al., 2024) and GSM8K (Cobbe et al., 2021). For general reasoning, we employ ARC-Challenge (Clark et al., 2018) and BBH (Suzgun et al., 2023). All benchmarks are evaluated under a zero-shot CoT setting, except for BBH, which uses 3-shot CoT. For BBH, we choose 15 subjects where the models exhibit comparable performance. Further details, including prompt templates and selected BBH subjects, are provided in Appendix A.

**Baselines.** We apply our method to two recent SOTA probability-level ensemble methods: GaC (Yu et al., 2024) and UniTE (Yao et al., 2025). GaC performs ensembling only when the main LLM's next-token probability falls below 0.5. In contrast, UniTE represents the SOTA among methods that ensemble at every generation step. In our setting, "**X + SAFE**" means that the ensemble method **X** is applied only at the token positions that SAFE identifies as requiring ensembling.

| Method | MMLU-redux | | MATH500 | | GSM8K | | BBH | | ARC-C | | Avg. |
|---|---|---|---|---|---|---|---|---|---|---|---|
| | Accuracy | E/T | Accuracy | E/T | Accuracy | E/T | Accuracy | E/T | Accuracy | E/T | Accuracy |
| Internlm3-8B | 76.89 | - | 74.8 | - | 90.14 | - | 82.26 | - | 90.27 | - | 82.87 |
| Qwen2.5-7B | 74.88 | - | 72.4 | - | 91.81 | - | 79.15 | - | 88.74 | - | 81.40 |
| EXAONE3.5-7.8B | 73.25 | - | 72.8 | - | 90.45 | - | 78.75 | - | 90.44 | - | 81.14 |
| *Two-model ensembling (Internlm3 + Qwen2.5)* | | | | | | | | | | | |
| GaC | 77.00 (+0.11) | 8.43 | 74.2 (-0.6) | 1.04 | 91.28 (-0.53) | 0.82 | 82.34 (+0.08) | 5.69 | 90.61 (+0.34) | 10.22 | 83.09 (+0.22) |
| **GaC + SAFE** | 77.11 (+0.22) | 5.23 | 76.0 (+1.2) | 0.71 | 91.36 (-0.45) | 0.67 | 82.34 (+0.08) | 3.73 | **91.13** (+0.86) | 6.22 | 83.59 (+0.72) |
| UniTE | 73.39 (-3.5) | 100 | 59.6 (-15.2) | 100 | 75.06 (-16.75) | 100 | 79.58 (-2.68) | 100 | 87.97 (-2.30) | 100 | 75.12 (-7.75) |
| **UniTE + SAFE** | **77.81** (+0.92) | 12.59 | **77.4** (+2.6) | 3.82 | **92.04** (+0.23) | 5.16 | **82.97** (+0.71) | 10.35 | 90.78 (+0.51) | 14.47 | **84.20** (+1.33) |
| *Two-model ensembling (Qwen2.5 + EXAONE3.5)* | | | | | | | | | | | |
| GaC | 76.01 (+1.13) | 13.42 | 75.4 (+2.6) | 2.31 | 92.65 (+0.84) | 2.60 | 79.61 (+0.46) | 8.15 | 90.27 (-0.17) | 14.66 | 82.79 (+1.39) |
| **GaC + SAFE** | **76.79** (+1.91) | 7.52 | 76.4 (+3.6) | 1.09 | 92.57 (+0.76) | 1.26 | 79.66 (+0.51) | 4.51 | 90.78 (+0.34) | 8.31 | 83.24 (+1.84) |
| UniTE | 53.75 (-21.13) | 100 | 43.4 (-29.4) | 100 | 77.03 (-14.78) | 100 | 67.45 (-11.70) | 100 | 72.61 (-17.83) | 100 | 62.85 (-18.55) |
| **UniTE + SAFE** | 76.54 (+1.66) | 17.24 | **76.4** (+3.6) | 4.69 | **92.72** (+0.91) | 5.60 | **81.69** (+2.54) | 14.03 | **90.78** (+0.34) | 19.24 | **83.63** (+2.23) |
| *Two-model ensembling (Internlm3 + EXAONE3.5)* | | | | | | | | | | | |
| GaC | 76.36 (-0.53) | 8.71 | 75.8 (+1.0) | 1.14 | **90.75** (+0.30) | 0.88 | **81.57** (-0.69) | 6.32 | 90.78 (+0.34) | 10.07 | 83.05 (+0.18) |
| **GaC + SAFE** | 77.21 (+0.32) | 5.94 | 77.2 (+2.4) | 0.84 | 90.67 (+0.22) | 0.72 | 81.54 (-0.72) | 4.38 | **91.72** (+1.28) | 6.92 | **83.67** (+0.80) |
| UniTE | 72.51 (-4.38) | 100 | 73.6 (-1.2) | 100 | 89.31 (-1.14) | 100 | 78.04 (-4.22) | 100 | 88.23 (-2.21) | 100 | 80.34 (-2.53) |
| **UniTE + SAFE** | 76.08 (-0.81) | 15.84 | 77.0 (+2.2) | 4.72 | **90.75** (+0.30) | 5.55 | 81.37 (-0.89) | 13.75 | 90.27 (-0.17) | 17.89 | 83.09 (+0.22) |
| *Three-model ensembling (Internlm3 + Qwen2.5 + EXAONE3.5)* | | | | | | | | | | | |
| UniTE | 73.92 (-2.97) | 100 | 76.0 (+1.2) | 100 | 91.28 (-0.53) | 100 | 77.47 (-4.79) | 100 | 87.20 (-3.24) | 100 | 81.17 (-1.70) |
| **UniTE + SAFE** | **77.60** (+0.71) | 16.18 | **79.0** (+4.2) | 4.12 | **92.04** (+0.23) | 5.14 | **82.77** (+0.51) | 12.74 | **91.55** (+1.11) | 18.60 | **84.59** (+1.72) |

Table 2: Ensembling results of models with substantially different tokenizations using CoT. E/T (%) represents the percentage of ensembling during generation, computed as $\frac{\#\mathbf{Ensemble}}{\#\mathbf{Token}}$ (%). Numbers in parentheses denote the performance gap relative to the best-performing individual model.

## 4.2 Implementation

**KV Cache Implementation.** KV caching is essential for efficient generation in LLMs. However, unlike standard generation settings where previously generated tokens remain fixed, ensemble generation may replace tokens during the ensembling process, leading to inconsistencies between the cache and the actual input sequence. Consequently, prior approaches have typically avoided implementing KV cache management, leaving it as future work. In contrast, our method updates each model's KV cache at the end of every ensemble step to align with the ensembled output, and uses this updated cache in the next step, thereby ensuring cache consistency. We apply our KV cache management to all baselines in our experiments. Please refer to Appendix D for details.

**Hardware and Hyperparameters.** We configure our method as follows. Following the approach of UniTE (Yao et al., 2025) for selecting a primary model, we select the model with the best average performance as $M_{\mathrm{draft}}$. For probability sharpening, we apply the heuristic strategy in our main results with the threshold $\lambda$ set to 0.1. The drafter generates tokens in chunks of 5, and all models use greedy decoding with a maximum output length of 2048. For ensembling, each model is loaded onto a separate GPU, with all experiments conducted on RTX 3090 GPUs with FP16 precision and FlashAttention-2 (Dao, 2023) enabled.

## 4.3 Main Analysis

Results of ensembling models with substantially different tokenization schemes are shown in Table 2. We also report E/T, the percentage of tokens that undergo ensembling over the entire sequence.

**SAFE improves performance with less ensembling.** Overall, SAFE generally outperforms individual models, making existing ensemble methods practical even under CoT. As shown in Table 2, the baseline UniTE struggles significantly under a CoT setting, consistently underperforming individual models across all experiments. This is because it ensembles at every generation step, which increases the frequency of OOV-like tokens and consequently corrupts probability distributions. In contrast, applying SAFE enables UniTE to achieve the best performance in many cases (9/15) while reducing the ensemble frequency (E/T) to fewer than 20% of tokens. This highlights the importance of determining when to ensemble, particularly when generating long sequences with models that use heterogeneous tokenizers. GaC, on the other hand, is more robust, since it performs ensembling only when the main LLM's probability falls below 0.5, yielding an unintended but beneficial effect of preventing the introduction of OOV-like tokens. Nevertheless, SAFE further improves GaC's performance while reducing the number of ensemble operations.

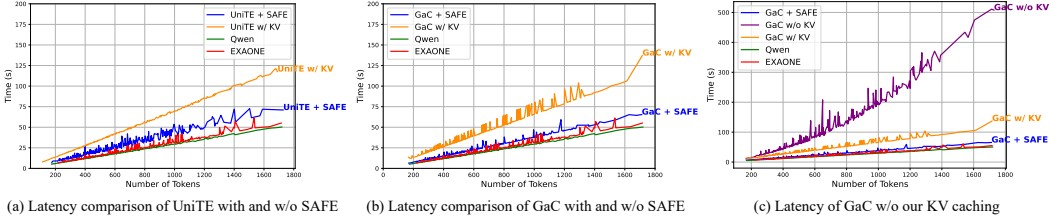

(a) Latency comparison of UniTE with and w/o SAFE    (b) Latency comparison of GaC with and w/o SAFE    (c) Latency of GaC w/o our KV caching

Figure 5: Latency comparison on MATH500. Our method shows similar latency compared to individual models, even when generating long sequences. *w/ KV* indicates that our KV caching strategy is applied. Note that the time-axis scale in (c) differs from (a) and (b).

One interesting finding is that much less ensembling is required in math datasets. When SAFE is applied with UniTE, ensembling is triggered for only 4.85% of tokens on average in math datasets, whereas it rises to 15.24% in general-domain datasets, which is nearly three times higher. We attribute this to the nature of math responses, which often contain equations or structured expressions with limited variation, leading to higher agreement among verifier models. In contrast, responses in general-domain datasets allow for greater linguistic variability, which reduces agreement across models and thus requires more frequent ensembling.

**Increasing the number of models is not always optimal.** As shown in Table 2, ensembling three models does not consistently outperform two-model ensembling, whereas ensembling the top-2 best performing models typically yields the strongest results. This suggests that when model rankings are known, restricting ensembling to the top-2 models is both effective and efficient. On the other hand, when rankings are unknown, ensembling multiple comparable models provides stable, though not necessarily optimal, performance.

**SAFE can be as fast as individual models.** A key challenge in LLM ensembling is achieving inference speed comparable to running a single model. As shown in Figure 5, SAFE closely matches the latency of individual models when generating hundreds of tokens, regardless of the underlying ensemble method. Moreover, under the same computational resources, it significantly improves efficiency over existing ensemble methods when generating long responses. This efficiency stems from three properties of our approach. First, only $M_{\text{draft}}$ is responsible for autoregressive generation. Second, SAFE substantially reduces the number of ensembling. Third, our KV caching strategy further improves efficiency, as illustrated in Figure 5(c). We provide more comparisons in Appendix E.

**SAFE further improves performance when ensembling models with similar tokenization.** We further evaluate SAFE on models with highly similar tokenizations, where more than 99% of Oxford 5000 words are tokenized identically. As shown in Table 3, the performance drop of existing methods is less severe than in Table 2. This is because highly aligned tokenization schemes greatly reduce the occurrence of OOV-like tokens, leading to more stable ensembling. Nevertheless, applying SAFE to existing methods consistently improves performance, yielding over a 9% gain on MATH500 compared to the best-performing individual model.

### 4.4 ABLATION STUDY

We conduct ablation studies on probability sharpening strategy and the drafter's sequence length.

| Method | MMLU-redux | MATH500 | GSM8K |
|---|---|---|---|
| Qwen2-7B | 69.25 | 49.8 | 85.90 |
| Llama3.1-8B | 68.51 | 47.6 | 82.56 |
| *Two-model ensembling (Qwen2 + Llama3.1)* | | | |
| GaC | 69.50 (+0.25) | 52.4 (+2.6) | 85.37 (-0.53) |
| **GaC + SAFE** | **69.99** (+0.74) | **59.4** (+9.6) | **86.66** (+0.76) |
| UniTE | 68.90 (-0.35) | 54.0 (+4.2) | 79.98 (-5.92) |
| **UniTE + SAFE** | **69.71** (+0.46) | **55.6** (+5.8) | **84.08** (-1.82) |

Table 3: Ensembling results of models with similar tokenization.

| Method | MMLU-redux | MATH500 | GSM8K |
|---|---|---|---|
| Internlm3-8B | 76.89 | 74.8 | 90.14 |
| Qwen2.5-7B | 74.88 | 72.4 | 91.81 |
| *Two-model ensembling (Internlm3 + Qwen2.5)* | | | |
| GaC | 77.00 | 74.2 | 91.28 |
| GaC + SAFE (w/o sharpen.) | 77.11 | 75.2 | **91.36** |
| **GaC + SAFE** | **77.11** | **76.0** | **91.36** |
| UniTE | 73.39 | 59.6 | 75.06 |
| UniTE + SAFE (w/o sharpen.) | 77.53 | 76.6 | 91.66 |
| **UniTE + SAFE** | **77.81** | **77.4** | **92.04** |

Table 4: Ablation on probability sharpening strategy.

| Sharpening Method | MMLU-redux | MATH500 | GSM8K | Avg. |
|---|---|---|---|---|
| UniTE + SAFE (no sharpening) | 77.53 | 76.6 | 91.66 | 81.93 |
| UniTE + SAFE ($\lambda = 0.1$) | 77.81 | 77.4 | 92.04 | 82.42 |
| UniTE + SAFE ($\lambda = 0.2$) | 77.28 | 77.6 | 92.04 | 82.31 |
| UniTE + SAFE ($\lambda = 0.3$) | 77.21 | 76.6 | 91.89 | 81.90 |
| UniTE + SAFE (geometric mean) | **78.31** | **77.6** | **92.27** | **82.73** |

Table 5: Performance across different probability sharpening methods. UniTE + SAFE is used to ensemble two models, Internlm3-8B-Instruct and Qwen2.5-7B-Instruct.

| Method | MMLU-redux | GSM8K | ARC-C |
|---|---|---|---|
| Internlm3-8B | 76.89 | 90.14 | 90.27 |
| Qwen2.5-7B | 74.88 | 91.81 | 88.74 |
| *Two-model ensembling (Internlm3 + Qwen2.5)* | | | |
| UniTE + SAFE$_3$ | 77.67 | 91.66 | 90.19 |
| UniTE + SAFE$_5$ | 77.81 | **92.04** | **90.78** |
| UniTE + SAFE$_8$ | **78.31** | **92.04** | **90.78** |

Table 6: Ablation on drafter sequence length. SAFE$_n$ denotes generation of $n$-token sequences.

**Probability sharpening.** As shown in Table 4, incorporating probability sharpening consistently improves performance across benchmarks and ensemble methods. This result highlights that sharpening is beneficial for choosing an accurate token when the ensemble distribution becomes overly smooth. To further examine whether different sharpening strategies provide similar benefits, we conduct additional experiments by varying the threshold $\lambda$ in our heuristic sharpening method and by applying the geometric mean. As demonstrated in Table 5, multiple sharpening strategies offer consistent gains. The geometric mean generally yields strong performance, which we attribute to its ability to gather the probability mass dispersed across multiple tokens by individual models and concentrate it on the token with the highest consensus among the models. However, considering that the arithmetic mean is widely used and often required, our heuristic strategy remains useful in such settings. Therefore, we leave the specific choice of sharpening method as a flexible design decision, allowing each ensemble method to adopt the sharpening strategy most suitable for its characteristics. The core takeaway is the importance of mitigating an overly smooth ensemble distribution. Regarding the threshold $\lambda$ used in our heuristic sharpening strategy, the performance does not change drastically across different values. However, setting the threshold too high reduces the number of tokens subject to sharpening, which in turn diminishes its effectiveness.

**Drafter's sequence length.** Table 6 presents the ablation on drafter sequence length. Generating short sequences may fail to capture differences in tokenization across models, causing slight performance drops. Conversely, generating overly long sequences does not harm accuracy but may reduce efficiency, as shown in Figure 10. This is because longer sequences force the drafter to regenerate tokens more often from the ensembled token. A length of 5 provides the best balance between accuracy and efficiency.

## 5  CONCLUSION

In this paper, we examined probability-level ensemble methods for long-form generation and showed that deciding *when* to ensemble is critical for both accuracy and efficiency. To this end, we proposed **SAFE**, a generate-verify-ensemble framework that triggers ensembling only when safe and necessary, guided by tokenization mismatch and consensus in models' next-token probability distributions. SAFE further improved accuracy via probability sharpening to mitigate smooth ensemble distribution, and our KV cache implementation enabled much faster ensembling in long-form generation. Experiments demonstrated that SAFE outperforms existing methods with only a few ensemble operations across widely used 7B-scale model combinations. We believe SAFE offers a practical step toward making LLM ensembling both robust and deployable in real-world applications.

ACKNOWLEDGEMENTS

This work was partly supported by Institute for Information & communications Technology Planning & Evaluation (IITP) grants (No. RS-2019-II190075, Artificial Intelligence Graduate School Program (KAIST), No. RS-2024-00457882, AI Research Hub Project, No.2022-0-00713, Meta-learning applicable to real-world problems, No. 2022-0-00984, Development of Artificial Intelligence Technology for Personalized Plug-and-Play Explanation and Verification of Explanation) funded by the Korea government (MSIT).

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

## A    EXPERIMENTAL DETAILS

In this section, we provide additional experimental details. We first present the exact prompts and BBH subjects used in our experiments, followed by an explanation of baseline selection.

**Prompts** For multiple-choice questions, we follow the template in Figure 6. For math datasets, we use the template in Figure 7.

> Question: [question]
> A.  …
> B.  …
> C.  …
> D.  …
> Provide your step-by-step reasoning first, and then print "The answer is (X)" where X is the answer choice (one capital letter), at the end of your response.

Figure 6: Prompt template used for multiple-choice questions.

> [question]
> Please reason step by step, and put your final answer within \boxed{}.

Figure 7: Prompt template used for math problems.

**BBH subjects** In our main experiments (Table 2), we filter BBH subjects to include only subjects where models exhibit comparable performance. This choice is based on the observation of UniTE (Yao et al., 2025) that ensemble is meaningful when the base models exhibit similar performance levels. The chosen 15 BBH subjects are: boolean expressions, causal judgement, date understanding, disambiguation qa, formal fallacies, logical deduction three objects, movie recommendation, navigate, penguins in a table, reasoning about colored objects, ruin names, salient translation error detection, snarks, temporal sequences, and tracking shuffled objects three objects.

**Baseline selection** As our purpose is to enhance the stability and efficiency of *probability-level ensemble* methods, we focus on recent, state-of-the-art probability-level ensemble methods as baselines. Specifically, we consider two representative methods that differ in when ensembling is performed. The first is GaC, which ensembles only when the main LLM's probability falls below 0.5. To the best of our knowledge, this is the only work that does not ensemble at every token generation. The second is UniTE, which achieves state-of-the-art performance among methods that ensemble at every generation step. By applying SAFE to both methods, we demonstrate its versatility and its effectiveness in determining when to ensemble.

## B    LIMITATIONS AND FUTURE WORK

**SAFE** does not always guarantee superior performance compared to the best-performing individual model. As shown in Table 2, in a few cases SAFE performs slightly worse than the best-performing individual models. We believe this is because the most confident token in the ensemble distribution is not necessarily the optimal choice, as poorly performing models can distort the ensemble distribution by elevating an incorrect token as the most confident. Nevertheless, SAFE generally outperforms individual models by ensuring a stable ensemble distribution when applied to ensembles of models with similar capability.

Additionally, our experiments are limited to non-reasoning models. Extending SAFE to reasoning models (Yang et al., 2025; Guo et al., 2025) would be a promising direction, as reasoning models have recently gained significant attention.

## C    CORRECTNESS OF ENSEMBLE DISTRIBUTION VERIFICATION

Theorem 1 guarantees that applying the ensemble distribution verification criteria does not compromise accuracy.

**Theorem 1.** *Let $\mathbf{t}_{<j}$ denote the drafter's prefix. For each verifier model $LLM_v \in M_{\mathrm{ver}}$, let $t^v_{v_j}$ be its next token, aligned such that $\mathbf{t^v}_{<v_j}$ is the $LLM_v$'s tokenization of $\mathbf{t}_{<j}$. Let $P_v$ be the next-token probability distribution of $LLM_v$. Suppose (i) all verifier models unanimously agree on the next token $t^v_{v_j}$, where $t^v_{v_j} = \arg\max_t P_v(t \mid \mathbf{t^v}_{<j})$ for all $v$, or (ii) their average probability in $t^v_{v_j}$ exceeds $\frac{1}{2}$:*

$$\frac{1}{|M_{\mathrm{ver}} \cup M_{\mathrm{draft}}|} \sum_{v \in M_{\mathrm{ver}} \cup M_{\mathrm{draft}}} P_v(t^v_{v_j} \mid \mathbf{t^v}_{<v_j}) > \frac{1}{2}.$$

*Then $t_j$ is the token selected by the ensemble, i.e., $t_j = \arg\max_t P_{\mathrm{ens}}(t \mid \mathbf{t}_{<j})$, where $P_{\mathrm{ens}}$ is the ensemble distribution.*

*Proof.* Define the ensemble distribution as average of all models' distributions which is aligned with $M_{\mathrm{draft}}$'s tokenization

$$P_{\mathrm{ens}}(t_j \mid \mathbf{t}_{<j}) \triangleq \frac{1}{|M_{\mathrm{ver}}| + 1} \sum_{v \in M_{\mathrm{ver}} \cup M_{\mathrm{draft}}} P_v\left(t^v_{v_j} \mid \mathbf{t^v}_{<v_j}\right).$$

**(i) Unanimous consensus across verifiers.**    If for all $v \in M_{\mathrm{ver}}$ we have $t^v_{v_j} = \arg\max_t P_v(t_j \mid \mathbf{t^v}_{<v_j})$, then for any token $u$,

$$P_v(t^v_{v_j} \mid \mathbf{t^v}_{<v_j}) \geq P_v(u \mid \mathbf{t^v}_{<v_j}) \quad \text{for all } v.$$

Since $t^v_{v_j}$ is aligned with $t_j$ for all $v$, averaging over $v$ preserves the inequality:

$$P_{\mathrm{ens}}(t_j \mid \mathbf{t}_{<j}) = \frac{1}{|M_{\mathrm{ver}}| + 1} \sum_v P_v(t^v_{v_j} \mid \mathbf{t^v}_{<v_j}) \geq \frac{1}{|M_{\mathrm{ver}}| + 1} \sum_v P_v(u \mid \mathbf{t^v}_{<v_j}) = P_{\mathrm{ens}}(u \mid \mathbf{t}_{<j}).$$

Hence, $t_j = \arg\max_t P_{\mathrm{ens}}(t \mid \mathbf{t}_{<j})$.

**(ii) Average probability above one half.**    Assume

$$P_{\mathrm{ens}}(t_j \mid \mathbf{t}_{<j}) = \frac{1}{|M_{\mathrm{ver}}| + 1} \sum_{v \in M_{\mathrm{ver}} \cup M_{\mathrm{draft}}} P_v(t^v_{v_j} \mid \mathbf{t^v}_{<v_j}) > \tfrac{1}{2}.$$

Because $P_{\mathrm{ens}}(\cdot \mid \mathbf{t}_{<j})$ is a probability distribution, $\sum_t P_{\mathrm{ens}}(t \mid \mathbf{t}_{<j}) = 1$. Thus, no other token $u \neq t_j$ can have $P_{\mathrm{ens}}(u \mid \mathbf{t}_{<j}) \geq P_{\mathrm{ens}}(t_j \mid \mathbf{t}_{<j})$, since two distinct tokens cannot both exceed $1/2$. Therefore, $t_j = \arg\max_t P_{\mathrm{ens}}(t \mid \mathbf{t}_{<j})$.

In either case (i) or (ii), the ensemble selects $t_j$, proving the claim.    $\square$

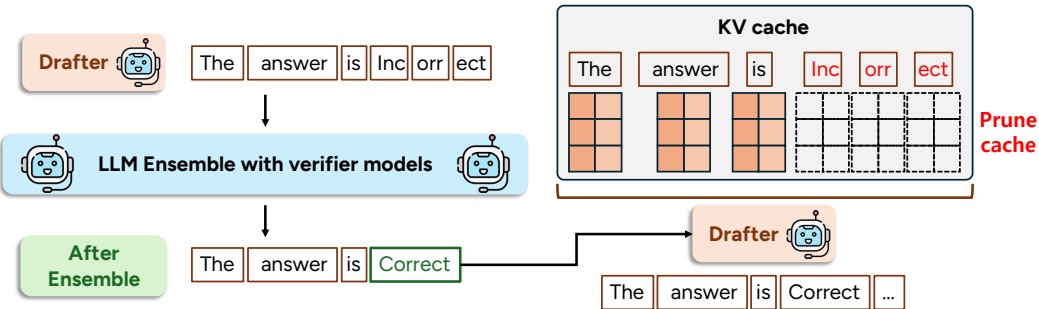

Figure 8: Our KV cache management. The cache is pruned to ensure alignment with the ensembled output.

## D    DETAILS OF OUR KV CACHE MANAGEMENT

While KV caching is essential for generating long responses, it introduces a challenge in ensemble settings because the token selected by the ensemble may differ from the tokens generated by individual models. As illustrated in Figure 8, the drafter initially generates `The answer is Incorrect`, but after ensembling the output becomes `The answer is Correct`. In this case, the drafter's KV cache, which contains states for the discarded token `Incorrect`, must be updated. To resolve this inconsistency, we prune each participating model's KV cache by a fixed buffer at the end of every ensemble step. This pruning ensures that each model's KV cache is consistent with the actual input sequences before producing the next token.

## E    LATENCY COMPARISON

We present additional latency comparisons on general-domain datasets, where the proportion of ensembled tokens is higher than in math datasets. Figure 9 shows the latency of SAFE on MMLU-redux dataset. As illustrated, SAFE substantially reduces latency compared to existing ensemble methods and achieves speeds comparable to individual models when generating hundreds of tokens, even in general-domain tasks. These results highlight the practical applicability of SAFE, demonstrating that it enables efficient ensembling across diverse domains.

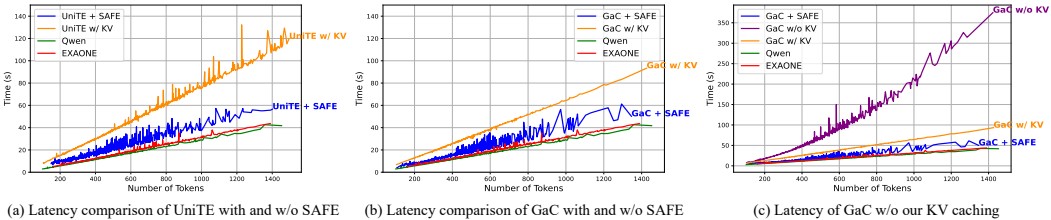

(a) Latency comparison of UniTE with and w/o SAFE    (b) Latency comparison of GaC with and w/o SAFE    (c) Latency of GaC w/o our KV caching

Figure 9: Latency comparison on MMLU-redux. SAFE significantly improves efficiency on general-domain tasks. Note that the time-axis scale in (c) differs from (a) and (b).

## F    ABLATION ON THE DRAFTER'S SEQUENCE LENGTH

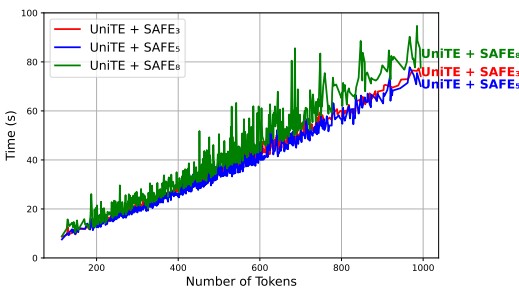

Figure 10: Latency comparison depending on the drafter's sequence length, where $n$ denotes the drafter's sequence length in $\text{SAFE}_n$. Generating a longer sequence reduces efficiency. We use MMLU-redux for the comparison.

## G    ENSEMBLING LARGER MODELS

Table 7 presents the results for ensembling 32B-scale models: Qwen2.5-32B-Instruct (Qwen et al., 2025) and EXAONE-3.5-32B-Instruct (An et al., 2024), evaluated on MMLU-redux and MATH500. For MMLU-redux, we report two variants: MMLU-redux$^*$ and the full MMLU-redux. MMLU-redux$^*$ includes only 21 subjects, excluding 9 subjects where Qwen2.5-32B-Instruct largely outperforms EXAONE-3.5-32B-Instruct by more than 10%, making ensembling less meaningful. As

shown in the table, applying SAFE to existing ensemble methods consistently outperforms the baselines, demonstrating its effectiveness on larger-scale models. The 9 subjects excluded in MMLU-redux* are: college chemistry, college mathematics, college physics, formal logic, electrical engineering, high school chemistry, professional accounting, clinical knowledge, and econometrics.

| Method | MMLU-redux* | MMLU-redux | MATH500 |
|---|---|---|---|
| | Accuracy | Accuracy | Accuracy |
| Qwen2.5-32B | 85.06 | **84.54** | 80.8 |
| EXAONE3.5-32B | 82.34 | 79.26 | 77.2 |
| *Two-model ensembling (Qwen2.5 + EXAONE3.5)* | | | |
| GaC | 84.70 | 82.73 | 80.4 |
| **GaC + SAFE** | **85.11** | 83.79 | **81.6** |

Table 7: Results of ensembling 32B-scale models.

## H  RELAXED ACCEPTANCE THRESHOLD

In the Ensemble Distribution Verification step, the acceptance threshold can be relaxed to support various scenarios, such as sampling or weighted combinations of probability distributions, rather than simple averaging. First, if the ensemble distribution is defined as a weighted sum of the models' next-token probability distributions, the second condition of Ensemble Distribution Verification becomes:

- **(Average probability above one half)**
  If $\sum_{LLM_v \in M_{\text{ver}} \cup M_{\text{draft}}} w_v P_v(t^v_{v_j} \mid \mathbf{t^v}_{<v_j}) > \frac{1}{2}$, we skip ensembling at $t_j$,

where $w_v$ is the weight assigned to $LLM_v$ and $\sum_v w_v = 1$.

If we wish to use sampling rather than greedy decoding, we can incorporate speculative sampling (Leviathan et al., 2023) into our framework. In this case, we do not use the first condition (Unanimous consensus among verifiers), and the second condition is relaxed as follows:

- (Average probability above one half) → **(Speculative sampling)**
  We skip ensembling at $t_j$ with probability $\min(1, \frac{P_{\text{ens}}(t_j|\mathbf{t}_{<j})}{P_{\text{draft}}(t_j|\mathbf{t}_{<j})})$.

Similar to greedy decoding setup, $P_{\text{ens}}(t_j|\mathbf{t}_{<j})$ is derived as:

$$P_{\text{ens}}(t_j|\mathbf{t}_{<j}) = \frac{1}{|M_{\text{ver}} \cup M_{\text{draft}}|} \sum_{LLM_v \in M_{\text{ver}} \cup M_{\text{draft}}} P_v(t^v_{v_j} \mid \mathbf{t^v}_{<v_j}),$$

where $t^v_{v_j}$ is the token corresponding to $t_j$ under $LLM_v$'s tokenization.

## I  FURTHER DISCUSSION OF OOV-LIKE TOKENS

In this section, we provide a more detailed explanation about OOV-like tokens, which we use to determine whether tokenization mismatch occurs. OOV-like tokens are defined as tokens that can corrupt the next-token probability distribution of a participating model due to differences in tokenization across models. As described in Section 3.2, whether a drafter token is classified as OOV-like is determined by checking whether the token boundary up to that token aligns with the tokenization boundaries of the verifiers.

One similar concept to OOV-like token is non-canonical tokenization, which refers to any tokenization other than the canonical tokenization for a given model. Several studies (Cao & Rimell, 2021; Geh et al., 2024; Chatzi et al., 2025; Vieira et al., 2025) have explored the use of non-canonical tokenizations in generation within a single LLM. However, we highlight two important distinctions between OOV-like tokens and non-canonical tokenizations. First, OOV-like tokens are defined

strictly at the token level. For example, consider the sequence "Hello, world". If (`Hello`, `_world`) is the canonical tokenization, then alternative tokenizations such as (`Hell`, `o`, `_world`) or (`Hello`, `_w`, `orld`) would be considered non-canonical tokenizations. However, we do not treat an entire tokenization like (`Hell`, `o`, `_world`) as an OOV-like case. Instead, we identify only the specific mismatched tokens within such non-canonical tokenizations, such as `Hell` or `_w`, as OOV-like tokens. Second, OOV-like tokens consider only the tokenizations used by the participating models, whereas non-canonical tokenization is a broader concept that encompasses all possible tokenizations of a sequence. For example, if the participating models tokenize "Hello" in only one way, (`Hello`), then no OOV-like token exists. In contrast, non-canonical tokenization includes any tokenization that can compose Hello, such as (`H`, `e`, `l`, `l`, `o`) or (`He`, `ll`, `o`), regardless of whether it is produced by the models. These distinctions directly relate to how we handle tokenization mismatch. We only assess mismatch at the token level, using only tokenizations of participating models. These two distinctions clearly represent differences from prior work on non-canonical tokenization.

## J  COMPARISON WITH METHODS THAT ENSEMBLING AT A SPAN-LEVEL

We discuss how our method differs from methods that ensemble at a larger-granularity, such as span-level ensembling, in addressing tokenization mismatch. Recently, several approaches (Xu et al., 2025; Liu et al., 2025) have proposed ensembling at a larger granularity, such as a set of words, to bypass the tokenization mismatch problem when ensembling LLMs that use heterogeneous tokenizers. These methods avoid tokenization mismatch by not performing ensembling at the token level and instead ensemble only at the span level. As a result, ensembling cannot occur inside a span, thereby sacrificing token-level probability information. In contrast, our method addresses tokenization mismatch within token-level ensembling. Our method resolves OOV-like issues by identifying token-level positions in the drafter's tokens where no mismatch occurs, and only these positions are considered valid candidates for ensembling. This allows ensembling to take place within a span while avoiding OOV-like issues, providing greater flexibility in where ensembling can occur.

## K  QUALITATIVE RESULTS

In this section, we present qualitative results of SAFE. Tables 8 to 11 compare responses from existing ensemble methods with those obtained when SAFE is applied. As shown in Table 8, SAFE can successfully yield the correct solution even when individual models fail to produce the correct answer or reasoning. Tables 8 and 11 further illustrate how existing ensemble methods suffer from repeated generation of erroneous tokens due to the introduction of OOV-like tokens. Finally, Tables 9 and 10 highlight cases where existing ensemble methods generate incorrect reasoning that leads to wrong answers.

## L  LLM USAGE

We used the LLM solely for the purpose of polishing the writing.

| Question | Sofia and Tess will each randomly choose one of the 10 integers from 1 to 10. What is the probability that neither integer chosen will be the square of the other? (A) 0.64 (B) 0.72 (C) 0.81 (D) 0.95 |
|---|---|
| Answer | (D) 0.95 |
| Internlm3-8B-Instruct | To determine the probability that neither integer chosen by Sofia and Tess will be the square of the other, we can follow these steps: 1. Identify the squares of the integers from 1 to 10: The squares of the integers from 1 to 10 are: $[1^2 = 1, 2^2 = 4, 3^2 = 9]$ So, the squares are 1, 4, and 9. 2. Determine the total number of possible outcomes: Each of Sofia and Tess can choose any of the 10 integers, so the total number of possible outcomes is: $[10 \times 10 = 100]$ 3. Count the number of outcomes where one integer is the square of the other: We need to count the pairs $((a, b))$ where $(a)$ is the square of $(b)$ or $(b)$ is the square of $(a)$. The pairs are: $[(1, 1), (1, 4), (1, 9), (4, 1), (4, 4), (4, 9), (9, 1), (9, 4), (9, 9)]$ There are 9 such pairs. 4. Calculate the number of outcomes where neither integer is the square of the other: $[100 - 9 = 91]$ 5. Calculate the probability: $[\frac{91}{100} = 0.91]$ However, none of the provided answer choices match 0.91. ... The answer is C. |
| Qwen2.5-7B-Instruct | To solve this problem, we need to determine the probability that neither of the two chosen integers is the square of the other. First, let's identify the squares of integers from 1 to 10: $(1^2 = 1), (2^2 = 4), (3^2 = 9)$ So, the only pairs of numbers in the range from 1 to 10 where one number is the square of the other are:(1, 1), (4, 2), (4, 4), (9, 3), (9, 9) There are 5 such pairs. Since each person can independently choose any of the 10 integers, there are a total of $(10 \times 10 = 100)$ possible outcomes when both Sofia and Tess make their choices. Next, we calculate the number of outcomes where at least one of the chosen integers is the square of the other. This includes the 5 pairs listed above and their reverses (since if (4, 2) is a pair, so is (2, 4)). However, since (1, 1), (4, 4), and (9, 9) are symmetric, they are already counted once. Therefore, we have:(1, 1) - 1 way, (4, 2) and (2, 4) - 2 ways, (4, 4) - 1 way, (9, 3) and (3, 9) - 2 ways, (9, 9) - 1 way. This gives us a total of $(1 + 2 + 1 + 2 + 1 = 7)$ pairs where one number is the square of the other. Thus, the number of outcomes where neither integer is the square of the other is $(100 - 7 = 93)$. The probability that neither integer chosen will be the square of the other is therefore: $[\frac{93}{100} = 0.93]$ However, this does not match any of the provided options exactly. ... Given the options, the closest match to our calculated probability of 0.93 is 0.95. The answer is (D). |
| UniTE | To determine the probability that neither integer chosen by SofÃa and Tess will be the square of the other, we need to follow these steps: 1. Identify the squares of the integers from 1 to 10: The squares of the integers from 1 to 10 are: $[1^2 = 1, 2^2 = 4, 3^2 = 9]$ So, the squares are 1, 4, and 9. 2. Determine the total number of possible outcomes: Each of SofÃÃÃÃÃ... |
| UniTE + **SAFE** | To determine the probability that neither integer chosen by Sofia and Tess will be the square of the other, we can follow these steps: 1. Identify the squares of the integers from 1 to 10: The squares of the integers from 1 to 10 are: $[1^2 = 1, 2^2 = 4, 3^2 = 9]$ So, the squares are 1, 4, and 9. 2. Determine the total number of possible outcomes: Each of Sofia and Tess can choose any of the 10 integers. Therefore, the total number of possible outcomes is: $[10 \times 10 = 100]$ 3. Count the number of outcomes where one integer is the square of the other: We need to count the pairs $((a, b))$ where $(a)$ is the square of $(b)$ or $(b)$ is the square of $(a)$. The pairs are: $[(1, 1), (4, 2), (2, 4), (9, 3), (3, 9)]$ There are 5 such pairs. 4. Calculate the number of outcomes where neither integer is the square of the other: $[100 - 5 = 95]$ 5. Calculate the probability: $[\frac{95}{100} = 0.95]$ Therefore, the probability that neither integer chosen will be the square of the other is D. |

Table 8: Response comparisons of UniTE and UniTE + **SAFE**. Some verbose outputs are omitted due to space constraints.

| Question | Terry eats 2 yogurts a day. They are currently on sale at 4 yogurts for $5.00. How much does he spend on yogurt over 30 days? |
|---|---|
| Answer | 75 |
| Qwen2.5-7B-Instruct | Let's break down the problem step by step to find out how much Terry spends on yogurt over 30 days. 1. Determine the number of yogurts Terry eats in 30 days: Terry eats 2 yogurts per day. Over 30 days, he will eat $(2 \times 30 = 60)$ yogurts. 2. Calculate how many sets of 4 yogurts are in 60 yogurts: Since the yogurts are sold in sets of 4 for $5.00, we need to find out how many such sets are in 60 yogurts. The number of sets is $(\frac{60}{4} = 15)$. 3. Calculate the total cost: Each set of 4 yogurts costs $5.00. For 15 sets, the total cost is $(15 \times 5 = 75)$ dollars. Therefore, the amount Terry spends on yogurt over 30 days is $\boxed{75}$ |
| EXAONE-3.5-7.8B-Instruct | To determine how much Terry spends on yogurt over 30 days, let's break down the problem step by step: 1. Calculate Daily Yogurt Consumption: - Terry eats 2 yogurts per day. 2. Determine Total Yogurt Consumption Over 30 Days: Over 30 days, Terry will consume $(2 \text{ yogurts/day} \times 30 \text{ days} = 60 \text{ yogurts})$. 3. Understand the Sale Price: The yogurts are sold at a rate of 4 yogurts for $5.00. 4. Calculate the Cost Per Yogurt: The cost per yogurt is $(\frac{5.00 \text{ dollars}}{4 \text{ yogurts}} = 1.25 \text{ dollars/yogurt})$. 5. Calculate Total Cost for 60 Yogurts: If each yogurt costs $1.25, then for 60 yogurts, the total cost will be $(60 \text{ yogurts} \times 1.25 \text{ dollars/yogurt} = 75 \text{ dollars})$. Therefore, the total amount Terry spends on yogurt over 30 days is $\boxed{75}$ |
| UniTE | To determine how much Terry spends on yogurt over 30 days, we can break down the problem into several steps: 1. Calculate the daily yogurt consumption in terms of the sale unit: Terry eats 2 yogurts per day. The sale is 4 yogurts for $5.00. Therefore, Terry consumes $(\frac{2}{4} = 0.5)$ of a sale unit per day. 2. Determine the daily cost based on the sale price: Since 4 yogurts cost $5.00, 1 yogurt costs $(\frac{5.00}{4} = 1.25)$. - Therefore, 0.5 of a sale unit (which is equivalent to 2 yogurts) costs $(0.5 \times 1.25 = 0.625)$. So, Terry spends $0.625 per day on yogurt. 3. Calculate the total cost over 30 days: Terry spends $0.625 per day. Over 30 days, the total cost is $(0.625 \times 30 = 18.75)$. Therefore, the total amount Terry spends on yogurt over 30 days is $\boxed{18.75}$ |
| UniTE + **SAFE** | To determine how much Terry spends on yogurt over 30 days, we can break down the problem into several steps: 1. Calculate the daily yogurt consumption cost: Terry eats 2 yogurts per day. The sale price is 4 yogurts for $5.00. To find the cost per yogurt, we divide the total cost by the number of yogurts: Cost per yogurt = $\frac{5.00}{4}$ = 1.25 dollars per yogurt. Since Terry eats 2 yogurts per day, the daily cost is: Daily cost = $2 \times 1.25 = 2.50$ dollars 2. Calculate the total cost over 30 days: Terry spends $2.50 per day on yogurt. Over 30 days, the total cost is: Total cost = $2.50 \times 30 = 75.00$ dollars Therefore, the total amount Terry spends on yogurt over 30 days is $\boxed{75.00}$ |

Table 9: Response comparisons of UniTE and UniTE + **SAFE**.

| Question | Compute $99^2 + 99 + 1$ in your head. |
|---|---|
| Answer | 9901 |
| Internlm3-8B-Instruct | To compute $99^2 + 99 + 1$ in my head, I can use algebraic manipulation to simplify the expression. Let's start by rewriting the expression in a more convenient form. First, let's denote $x = 99$. Then the expression becomes: $x^2 + x + 1$. Notice that this expression can be rewritten using the identity for the square of a binomial: $x^2 + x + 1 = (x + 1)^2 - x$. Substituting $x = 99$ back into the expression, we get: $99^2 + 99 + 1 = (99 + 1)^2 - 99$. Simplifying inside the parentheses, we have: $99 + 1 = 100$. So the expression becomes: $100^2 - 99$. Now, we know that $100^2 = 10000$, so the expression simplifies to: $10000 - 99 = 9901$. Therefore, the value of $99^2 + 99 + 1$ is $\boxed{9901}$ |
| Qwen2.5-7B-Instruct | To compute $99^2 + 99 + 1$ in my head, I will break it down into simpler steps. First, let's calculate $99^2$: [$99^2 = (100-1)^2 = 100^2 - 2 \cdot 100 \cdot 1 + 1^2 = 10000 - 200 + 1 = 9801$.] Next, we add 99 to the result: $9801 + 99 = 9900$. Finally, we add 1 to the result: $9900 + 1 = 9901$. So, the final answer is $\boxed{9901}$ |
| GaC | To compute $99^2 + 99 + 1$ in my head, I can use algebraic manipulation to simplify the expression. Let's start by rewriting the expression in a more convenient form. First, let's denote $x = 99$. Then the expression becomes: $x^2 + x + 1$. Now, we need to compute $x^2 + x + 1$ for $x = 99$. Notice that $x^2 + x + 1$ can be rewritten using the identity for the sum of cubes: $x^2 + x + 1 = (x + 1)^2 - x$. However, a simpler approach is to recognize that $x^2 + x + 1$ can be expressed as: $x^2 + x + 1 = (x+1)^2 - x + x = (x+1)^2$. So, we have: $99^2 + 99 + 1 = (99+1)^2 = 100^2$. Now, we just need to compute $100^2$: $100^2 = 10000$. Therefore, the value of $99^2 + 99 + 1$ is $\boxed{10000}$ |
| GaC + **SAFE** | To compute $99^2 + 99 + 1$ in my head, I can use algebraic manipulation to simplify the expression. Let's start by rewriting the expression in a more convenient form. First, let's denote $x = 99$. Then the expression becomes: $x^2 + x + 1$. Now, we can use the identity for the square of a binomial to rewrite $x^2 + x + 1$. Notice that: $x^2 + x + 1 = (x + 1)^2 - x$. Substituting $x = 99$ back into the expression, we get:$99^2 + 99 + 1 = (99 + 1)^2 - 99$. Simplifying inside the parentheses, we have: $99 + 1 = 100$. So the expression becomes: $100^2 - 99$. Next, we compute $100^2$: $100^2 = 10000$. Now, we subtract 99 from 10000: $10000 - 99 = 9901$. Therefore, the value of $99^2 + 99 + 1$ is $\boxed{9901}$ |

Table 10: Response comparisons of GaC and GaC + **SAFE**.

| Question | At 25°C, the maximum amount of $PbI_2$ that can be dissolved in 1.00 L of pure water is 1.0 mmol. Assuming complete dissociation, the solubility product, $K_{sp}$, for lead iodide at 25°C is (A) $1.0 \times 10^{-3}$, (B) $1.0 \times 10^{-6}$, (C) $1.0 \times 10^{-9}$, (D) $4.0 \times 10^{-9}$ |
|---|---|
| Answer | (D) |
| Qwen2.5-7B-Instruct | To solve this problem, we need to understand the relationship between the solubility of a compound and its solubility product constant $K_{sp}$. 1. Dissociation of Lead Iodide $PbI_2$: $PbI_2(s) \rightleftharpoons Pb^{2+}(aq) + 2I^-(aq)$ 2. Solubility in Water: Given that 1.0 mmol (or $1.0 \times 10^{-3} mol$) of $PbI_2$ dissolves in 1.00 L of pure water, we can determine the concentrations of the ions: The concentration of $Pb^{2+}$ is $1.0 \times 10^{-3}$M. The concentration of $I^-$ is $2 \times 1.0 \times 10^{-3} = 2.0 \times 10^{-3}$M. 3. Solubility Product Constant $K_{sp}$: The expression for the solubility product constant $K_{sp}$ for $PbI_2$ is: $K_{sp} = [Pb^{2+}][I^-]^2$ Substituting the concentrations: $K_{sp} = (1.0 \times 10^{-3})(2.0 \times 10^{-3})^2 = (1.0 \times 10^{-3})(4.0 \times 10^{-6}) = 4.0 \times 10^{-9}$ Therefore, the solubility product, $K_{sp}$, for lead iodide at 25°C is $4.0 \times 10^{-9}$. The answer is D. |
| EXAONE-3.5-7.8B-Instruct | To determine the solubility product constant $K_{sp}$ for lead iodide $PbI_2$, we need to follow these steps: 1. Understand the Dissociation Reaction: Lead iodide dissociates in water as follows: $PbI_2 \rightarrow Pb^{2+} + 2I^-$ 2. Determine the Concentrations: Given that the maximum amount of $PbI_2$ that can dissolve in 1.00 L of water is 1.0 mmol, we can find the molar concentrations of $Pb^{2+}$ and $I^-$ ions produced. Since 1 mmol of $PbI_2$ dissociates into 1 mmol of $Pb^{2+}$ and 2 mmol of $I^-$: $[Pb^{2+}] = \frac{1.0 \text{ mmol}}{1.0 \text{ L}} = 1.0$ mmol/L $= 1.0 \times 10^{-3}$ M, $[I^-] = \frac{2 \times 1.0 \text{ mmol}}{1.0 \text{ L}} = 2.0$ mmol/L $= 2.0 \times 10^{-3}$ M 3. Write the Expression for $K_{sp}$: The solubility product constant expression for $PbI_2$ is: $K_{sp} = [Pb^{2+}] \cdot [I^-]^2$ 4. Substitute the Concentrations into the $K_{sp}$ Expression: $K_{sp} = (1.0 \times 10^{-3}) \cdot (2.0 \times 10^{-3})^2 = (1.0 \times 10^{-3}) \cdot (4.0 \times 10^{-6}) = 4.0 \times 10^{-9}$ Therefore, the solubility product constant $K_{sp}$ for lead iodide $PbI_2$ at 25°C is $4.0 \times 10^{-9}$. The answer is (D). |
| GaC | To solve this problem, we need to understand the relationship between the solubility product constant $K_{sp}$ and the solubility of a compound. 1. Identifying the Dissociation Equation: Lead(II) iodide $PbI_2$ dissociates in water as follows: $PbI_2(s) \rightleftharpoons Pb^{2+}(aq) + 2I^-(aq)$ 2. Expressing Solubility in Terms of Concentrations: Let $s$ be the molar solubility of $PbI_2$. This means that when $PbI_2$ dissolves, it produces $s$ moles of PbÂ²âġ° per liter of solution and $2s$ moles of Iâââ... |
| GaC + **SAFE** | To solve this problem, we need to understand the relationship between the solubility product constant $K_{sp}$ and the solubility of a compound. For lead iodide $PbI_2$, the dissociation in water can be represented as: $PbI_2 \rightleftharpoons Pb^{2+} + 2I^-$ Given that the maximum amount of $PbI_2$ that can be dissolved in 1.00 L of pure water is 1.0 mmol (which is $1.0 \times 10^{-3}$ mol, we can determine the concentrations of the ions in solution. 1. Determine the concentration of $Pb^{2+}$ ions: Since 1.0 mmol of $PbI_2$ dissociates into 1.0 mmol of $Pb^{2+}$ ions, the concentration of $Pb^{2+}$ ions is: $[Pb^{2+}] = 1.0 \times 10^{-3}$M 2. Determine the concentration of $I^-$ ions: Each mole of $PbI_2$ dissociates into 2 moles of $I^-$ ions. Therefore, the concentration of $I^-$ ions is: $[I^-] = 2 \times 1.0 \times 10^{-3}$M $= 2.0 \times 10^{-3}$M 3. Calculate the solubility product constant $K_{sp}$: The expression for $K_{sp}$ for $PbI_2$ is: $K_{sp} = [Pb^{2+}][I^-]^2$. Substituting the concentrations we found: $K_{sp} = (1.0 \times 10^{-3})(2.0 \times 10^{-3})^2 = (1.0 \times 10^{-3})(4.0 \times 10^{-6}) = 4.0 \times 10^{-9}$ Therefore, the solubility product, $K_{sp}$, for lead iodide at 25°C is $4.0 \times 10^{-9}$. The answer is D. |

Table 11: Response comparisons of GaC and GaC + **SAFE**. Some verbose outputs are omitted due to space constraints.

