# OpenReview forum: "When to Ensemble: Identifying Token-Level Points for Stable and Fast LLM Ensembling"
_ICLR.cc/2026/Conference — ICLR 2026 Poster_

### Official Review · Reviewer_vm3X · 2025-10-23

**Soundness:** 3
**Presentation:** 2
**Contribution:** 3
**Rating:** 4
**Confidence:** 5

**Summary:**

This paper studies ensemble methods in long-form generation tasks, demonstrating that existing ensemble methods fail to perform well in these tasks because of tokenization mismatch across models and divergence in their next-token probability distributions. To address this problem, they propose SAFE, a novel ensemble method. Experiments show that SAFE achieves better performance and efficiency.

**Strengths:**

1. The focus on long generation/CoT tasks is a novel perspective for model ensemble research.
2. The proposed approach leads to better performance across different benchmarks and ensemble strategies. It also address the huge inference overhead.
3. The writing is clear and easy to follow.

**Weaknesses:**

1. Some related works should be discussed and compared, which could impact the novelty and relative performance:
    a) Speculative decoding is already utilized to accelerate model ensemble in CoS [1].

    a) The problem of OOV can be addressed by ensembling at a larger granularity (e.g., word of span), as shown in SweetSpan [2]. I think the same idea is used in SAFE.

    b) EVA [3] (which is the first token-level ensembling approach) shows that the divergence between models could be addressed by dynamically adjusting the weight of different models according to their confidence. I would suggest that the authors try these approaches.

2. For the verification step, the proposed token is only accepted when it is the most possible one for all models. However, modern LLMs typically use random sampling with topk/topp renorm for diversity. This is crucial for generating long sequences without rotting and further test-time scaling. I think expanding the acceptance threshold would be a more reasonable approach.

[1] Fu, Jiale, et al. "Fast Large Language Model Collaborative Decoding via Speculation."

[2] Xu, Yangyifan, et al. "Hit the sweet spot! span-level ensemble for large language models."

[3] Xu, Yangyifan, et al. "Bridging the Gap between Different Vocabularies for LLM Ensemble."

**Questions:**

1. What is the main contribution compared to existing approaches?
2. What would be the result if greedy decoding is replaced with common sampling techniques?

---

> ### Author Response · Authors · 2025-11-20
> **Response to Reviewer vm3X (1)**
>
> Dear Reviewer vm3X,
>
> We sincerely appreciate the reviewer’s valuable feedback. Your feedback has been instrumental in improving the clarity and quality of our work. We are also grateful for your positive remarks on our novel perspective for ensemble research, the strong experimental results, and the clarity of writing. Below, we provide our responses to address the reviewer’s concerns.
>
> ---
>
> **[W1 + Q1: Main contribution of our paper]**
>
> Thank you for the thoughtful suggestion, and we apologize for any ambiguity regarding the main contribution of our work. We have revised the manuscript to clearly articulate our contribution and incorporated the necessary discussions, including references and distinctions from existing work, in the revision.
> The main contribution of our paper is identifying a specific but practically important scenario in which probability-level LLM ensembling becomes problematic, and proposing a solution to address it. We find that, in the setting of ensembling models with different tokenizers for long-form generation, the positions at which ensembling occurs should be chosen carefully. To this end, we propose a method that determines appropriate token-level positions by jointly considering **tokenization mismatch** and **consensus in the models’ next-token probability distributions**.
> The problem setting we target and our corresponding solution introduce a new contribution that has not been addressed in prior works, either individually or in combination. Below, we explain our contribution by dividing it into the three perspectives mentioned by the reviewer.
>
> **W1-1: Comparison with CoS**
>
> The key difference is that we extend the speculative decoding framework to support collaboration among models with heterogeneous tokenizers. To achieve this, we propose an acceptance criterion guided by tokenization mismatch across models and consensus in their next-token probability distributions. Our main results show that addressing these two factors is critical for achieving strong performance in long-sequence generation. In contrast, CoS applies speculative decoding only to collaboration among models that share the same tokenizer, which does not consider divergence arising from tokenization mismatch. As a result, when ensembling models use different tokenizers, the drafter's tokens cannot be properly evaluated by other participating models due to tokenization misalignment, making CoS inapplicable in such scenarios. Since most open-source models use different tokenizers, we believe that extending speculative decoding to our problem setting is practically valuable. We have added a discussion of this point in Section 2.2 of the revised manuscript.
>
> **W1-2: Comparison with methods that ensembling at a larger granurality (e.g. word of span)**
>
> Our approach to addressing the OOV problem is fundamentally different from methods that ensemble at a larger granularity. Methods that ensemble at a larger level, such as span-level ensembling, simply mask the OOV-like issue by bypassing token-level operations. Consequently, they prevent ensembling from occurring within a span, thereby sacrificing token-level probability information.
>
> In contrast, our method resolves the OOV-like issue within token-level ensembling by identifying token-level positions in the drafter’s tokens where no tokenization mismatch occurs. This allows ensembling to take place within a span while avoiding the OOV-like issue, providing greater flexibility in where ensembling can occur compared with span-level approaches. For example, suppose each model produces the following spans:
>
> - Model 1: *1 + 1 = 3*
> - Model 2: *1 + 1 = 5*
>
> Span-level ensemble methods can only choose among these fixed span candidates, since each span is treated as an atomic unit.
>
> In contrast, our approach enables ensembling within a span. Specifically, Model 1 (as the drafter) first proposes *1 + 1 = 3*, after which Model 2 (as the verifier) evaluates this sequence. Both models would agree on the prefix *1 + 1 =*, and the final token can then be determined by ensembling their next-token probability distributions. This allows the ensemble to correct within a span and produce the correct output *1 + 1 = 2*. Our revised manuscript discusses this distinction in Appendix I.

---

> ### Author Response · Authors · 2025-11-20
> **Response to Reviewer vm3X (2)**
>
> **W1-3: Applying confidence-based weighting**
>
> We also considered dynamically adjusting ensemble weights according to each model’s confidence in our main results. GaC adopts the same principle used in EVA by adjusting ensemble weights based on a model’s confidence. As shown in Section 4.3, and as the reviewer noted, confidence-based weighting alone (e.g., GaC) is more robust than methods that do not adjust the weights at all (e.g., UniTE). Nevertheless, when SAFE is combined with these methods, we observe additional gains in both speed and performance.
>
> The reason is that confidence-based weighting determines each model’s contribution solely based on its confidence, without considering tokenization mismatch. As a result, a model may be excluded from ensembling simply because it has low confidence, even when no tokenization mismatch occurs. Conversely, a model may be included in ensembling despite having tokenization mismatch. In contrast, our method provides a more principled solution by explicitly identifying tokenization mismatches at the token level.
>
> Although we believe the GaC results already included in the paper sufficiently address the reviewer's concern, we also conducted additional experiments comparing the confidence-based weighting method proposed in EVA with SAFE. For the EVA’s weighting hyperparameters, we followed the configuration used in EVA for reasoning tasks. The results are presented in the table below. As shown, although confidence-based weighting helps mitigate OOV-like issues, SAFE is more effective overall.
>
> | Method | MATH500 | GSM8K |
> |-------------------|:--------------:|:--------:|
> | Internlm3-8B-Instruct | 74.8 |90.14 |
> | Qwen2.5-7B-Instruct | 72.4| 91.81 |
> | UniTE | 59.6 | 75.06 |
> | UniTE + EVA’s confidence-based weighting | 75.0 |88.32 |
> | **UniTE + SAFE** | **77.4** | **92.04** |
>
> ---
>
> **[W2 + Q2: Applying a relaxed acceptance threshold instead of greedy decoding]**
>
> We only reported greedy decoding results in the main experiments to remain consistent with prior ensemble studies, as most probability-level ensemble methods [1,2] adopt greedy decoding due to its strongest performance. However, our framework can naturally accommodate more flexible sampling strategies by relaxing the acceptance threshold.
>
> To do so, we apply speculative sampling [3], where the proposed token $t_{j}$ from the drafter is accepted with probability $\min(1, \frac{P_{\text{ens}}(t_{j}|\mathbf{t}\_{<j})}{P_{\text{draft}}(t_{j}|\mathbf{t}\_{<j})})$, where $P_{\text{ens}}$ and $P_{\text{draft}}$ denote the next-token probabilities under the ensemble distribution and the drafter, respectively. Similar to greedy decoding setup, $P_{\text{ens}}(t_{j}|\mathbf{t}_{<j})$ is derived as:
>
> $$
> P_{\text{ens}}(t_{j}|\mathbf{t}\_{<j})=\frac{1}{|M_\mathrm{ver}\cup M_\mathrm{draft}|}\sum_{LLM_v\in M_\mathrm{ver}\cup M_\mathrm{draft}} P_v(t_{v_j}^v\mid\mathbf{t^v}_{<v_j}),
> $$
>
> where $t_{v_j}^v$ is the token corresponding to $t_j$  under $LLM_v$’s tokenization, and $P_v$ denotes the probability distribution of $LLM_v$.
>
> The results for various sampling strategies are shown below. For topk sampling, we filter each model’s distribution to include only the topk tokens, and then apply the above acceptance threshold. We conducted these experiments using the ensemble of Internlm3-8B-Instruct and Qwen2.5-7B-Instruct.
>
> | Method | MMLU-redux | MATH500 | GSM8K |
> |-------------------|:--------------:|:--------:|:--------:|
> | UniTE + SAFE w/ greedy | 77.81 | 77.4 | 92.04 |
> | UniTE + SAFE w/ sampling (Temp.=1.0) | 76.57 |76.4 | 90.45 |
> | UniTE + SAFE w/ top10 | 77.18 | 76.8 | 91.13|
>
> As shown in the above table, our method readily supports generation strategies such as sampling by relaxing the acceptance threshold. Thank you for the valuable suggestion and we have added this discussion in Appendix G.
>
> [1] Yao, Yuxuan, et al. "Determine-then-ensemble: Necessity of top-k union for large language model ensembling." ICLR 2025.
>
> [2] Xu, Yangyifan, Jinliang Lu, and Jiajun Zhang. "Bridging the gap between different vocabularies for llm ensemble." NAACL 2024.
>
> [3] Leviathan, Yaniv, Matan Kalman, and Yossi Matias. "Fast inference from transformers via speculative decoding." ICML 2023.
>
> ---
>
> Once again, we sincerely thank you for your time and dedication. If you have any further questions or comments, please do not hesitate to reach out.

---

### Official Review · Reviewer_ux3H · 2025-10-26

**Soundness:** 3
**Presentation:** 3
**Contribution:** 3
**Rating:** 8
**Confidence:** 2

**Summary:**

This paper introduces a new framework, SAFE, to determine when to ensemble models' next-token generation probability. The framework allows ensemble happen only when (1) no tokenization mismatch occurs across models and (2) the models’ next-token distributions show low consensus. SAFE increase model inference efficiency and accuracy on long-form generation under ensemble.

**Strengths:**

Disclaimer: reviewer is not an expert on the topic.

1. The paper is well-presented.
2. I think the paper's idea of using token mismatching and token confidence to decide whether to ensemble is intuitive. SAFE can be applied to any existing next-token ensemble method.
3. The paper experiments clearly show gains for efficiency and accuracy for using SAFE.

**Weaknesses:**

Disclaimer: reviewer is not an expert on the topic.

I'd like to see an ablation study for how each of the two criteria in SAFE impact ensemble method performance.

I will check other reviewers' comment on the weakness part of this paper.

**Questions:**

N/A

---

> ### Author Response · Authors · 2025-11-20
> **Response to Reviewer ux3H**
>
> Dear Reviewer ux3H,
>
> We sincerely appreciate the reviewer’s valuable feedback. Your feedback has greatly contributed to enhancing the quality of our paper. We also thank you for your positive remarks on the clarity of writing, the soundness of our method, and the strong experimental results. Below, we provide our responses to address the reviewer’s concerns.
>
> ---
>
> **[W1: Ablation study of each component]**
>
> Thank you for providing valuable feedback that helps validate each component of SAFE. The table below shows the ablation results for the two criteria in SAFE when ensembling Internlm3-8B-Instruct and Qwen2.5-7B-Instruct. Applying OOV-like Token Verification clearly improves performance by preventing ensembling when OOV-like tokens are introduced. Further incorporating Ensemble Distribution Verification significantly reduces E/T, the percentage of tokens that undergo ensembling, which relates to efficiency.
>
> | **Method** | **MMLU-redux** | | **MATH500** | |
> |-------------------|:--------------:|:--------:|:---------:|:--------:|
> | | **Accuracy** | **E/T** | **Accuracy**| **E/T** |
> | Internlm3-8B-Instruct | 76.89| - | 74.8 | - |
> | Qwen2.5-7B-Instruct | 74.88| - | 72.4 | - |
> | UniTE | 73.39| 100 | 59.6 | 100 |
> | UniTE + SAFE (w/o ensemble distribution verification) | 77.35| 88.58 | 77.4 | 94.29 |
> | **UniTE + SAFE** | 77.81| 12.59 | 77.4 | 3.82 |
>
> ---
>
> Once again, thank you for your time and commitment in reviewing our paper, even though the topic may lie outside your expertise. We genuinely appreciate your effort, and we will do our best to address any remaining questions regarding our paper.

---

### Official Review · Reviewer_gPZv · 2025-10-28

**Soundness:** 3
**Presentation:** 3
**Contribution:** 2
**Rating:** 6
**Confidence:** 4

**Summary:**

The paper argues that LLM ensembling only needs to occur for a subset of tokens determined by: (1) the mismatch in tokenization across different models and (2) the level of consensus in their next-token probability distributions. The authors introduce SAFE (Stable And Fast LLM Ensembling), a framework designed to find the best tokens for ensembling during long-sequence generation. SAFE uses a speculative strategy where one "drafter" model generates a lookahead sequence of tokens, and the other "verifier" models identifies the specific token-level points within that sequence that require ensembling.

**Strengths:**

* The paper is well-motivated and clearly written.

* The experiments are fairly rigorous. The method achieves better quality and efficiency over baseline ensembling methods.

* A notable systems contribution is the implementation of KV caching for ensembling, which the authors also apply to baselines.

* It is an interesting finding that ensembling at a small fraction of tokens (less than 20%) can significantly improve the overall generation quality.

**Weaknesses:**

* The paper introduces the term "OOV-like," but this appears to be identical to the existing concept of "non-canonical" tokenization found in prior work [1, 2, 3, 4]. The paper fails to cite or discuss this relevant literature on tokenization.

* The probability sharpening strategy seems arbitrary. The choice of the 0.1 threshold is not well-justified. There are other standard ways to sharpen distributions (e.g., using a geometric mean instead of an arithmetic mean) that are not explored or compared.

* The experiments relating to efficiency could be more complete to show where exactly the speedups are coming from (see questions below).

[1] Cao, Kris, and Laura Rimell. "You should evaluate your language model on marginal likelihood over tokenisations." arXiv preprint arXiv:2109.02550 (2021).

[2] Geh, Renato Lui, et al. "Where is the signal in tokenization space?." arXiv preprint arXiv:2408.08541 (2024).

[3] Vieira, Tim, et al. "Language Models over Canonical Byte-Pair Encodings." arXiv preprint arXiv:2506.07956 (2025).

[4] Chatzi, Ivi, et al. "Canonical Autoregressive Generation." arXiv preprint arXiv:2506.06446 (2025).

**Questions:**

1. To confirm my understanding, is it true that if you apply SAFE ensembling between a model and itself, it will give the same result as just sampling directly from the model?  I want to confirm that ensembling does not induce temperature-scaling-like effects that boost performance.

2. The authors repeatedly claim that vocabulary alignment is a primary expense, e.g., "the number of costly ensemble operations grows with sequence length. This expense arises primarily from vocabulary alignment..." and "high computational cost of repeated vocabulary alignment makes such approaches inefficient." Can you elaborate on this? My understanding is that the primary bottleneck for ensembling is the cost of forward passes through multiple models, not vocabulary alignment. Why is alignment expensive in this setup?

3. The latency is shown to be lower, but what is the maximum memory usage (peak memory) of SAFE compared to the baselines?

4. What is the computational overhead of repeatedly calling the verifier models' tokenizers during the verify step?

---

> ### Author Response · Authors · 2025-11-20
> **Response to Reviewer gPZv (1)**
>
> Dear Reviewer gPZv,
>
> We sincerely appreciate the reviewer’s insightful feedback. Your comments have been invaluable in helping us strengthen our work from multiple perspectives. We are also grateful for your positive remarks on the clarity of writing, rigorous experiments, system contributions, and the findings of our paper. Below, we provide our responses to address the reviewer’s concerns.
>
> ---
> **[W1: Comparison with “non-canonical” tokenization]**
>
> Although OOV-like tokens share a similar high-level concept with non-canonical tokenization, there are two important distinctions.
>
> First, OOV-like tokens are defined strictly at the **token level**. For example, consider the sequence *“Hello, world”*. If *[Hello, _world]* is the canonical tokenization, then alternative tokenizations such as *[Hell, o, _world]* or *[Hello, _w, orld]* would be considered non-canonical tokenizations. However, we do not treat an entire tokenization like *[Hell, o, _world]* as an OOV-like case. Instead, we identify only the specific mismatched tokens within such non-canonical tokenizations, such as *“Hell”* or *“_w”,* as OOV-like tokens.
>
> Second, OOV-like tokens consider **only the tokenizations used by the models participating in the ensemble**, whereas non-canonical tokenization encompasses all possible tokenizations of a sequence. For example, if the participating models tokenize “Hello” in only one way, [Hello], then no OOV-like token exists. In contrast, non-canonical tokenization includes any tokenization that can compose Hello, such as [H, e, l, l, o] or [He, ll, o], regardless of whether it is produced by the models.
>
> These distinctions directly relate to how we handle tokenization mismatch. We only assess mismatch at the token level, using only tokenizations of participating models. We believe these two distinctions clearly represent differences from prior work on non-canonical tokenization.
>
> In addition to these differences, we did not discuss the cited papers because their problem setting differs substantially from ours. Our work focuses on **token-level collaboration across models with heterogeneous tokenizers**, whereas the cited works study how non-canonical tokenizations affect the generation behavior of a single model, without cross-model collaboration. We believed that comparing these studies with our paper could confuse readers regarding the objective of our work.
>
> Nevertheless, we fully agree with the reviewer that clarifying the distinction between non-canonical tokenization and our notion of OOV-like tokens would be helpful. Accordingly, we have added a discussion of these differences in Appendix H.
>
> ---
> **[W2: Regarding the probability sharpening strategy]**
>
> Thank you for the insightful feedback. Our probability sharpening strategy is intended as a small component of our overall method, and our goal was to illustrate that, in settings where models use different tokenizers, sharpening the probability distribution can help identify the most confident token during ensembling.
>
> To further examine whether different sharpening strategies provide similar benefits, we conduct additional experiments both by varying the threshold in our sharpening method and by applying alternative strategies such as a geometric mean. In these experiments, UniTE + SAFE is used to ensemble Internlm3-8B-Instruct and Qwen2.5-7B-Instruct. As shown in the table below, the improvement is not limited to a single sharpening technique, and multiple sharpening strategies offer consistent gains. This supports our motivation for introducing sharpening: an overly smooth ensemble distribution can hinder confident token selection. Based on the general benefit of introducing sharpening, we leave the specific choice of sharpening method as a flexible design decision and will include a detailed comparison of different sharpening methods. We believe that the core takeaway is the **importance of mitigating an overly smooth ensemble distribution**.
>
> Regarding the threshold used in our sharpening strategy, the performance does not change drastically across different threshold values. However, setting the threshold too high reduces the number of tokens subject to sharpening, which in turn diminishes its effectiveness.
>
> | **Sharpening Method**  | **MMLU-redux** | **MATH500** | **GSM8K** | **Avg.** |
> |----------------------------------|:----------:|:--------:|:------:|:------:|
> | UniTE + SAFE (no sharpening)  | 77.53   |  76.6  | 91.66  | 81.93 |
> | UniTE + SAFE (threshold = 0.1) | 77.81   |  77.4  | 92.04  | 82.42 |
> | UniTE + SAFE (threshold = 0.2) | 77.28   |  77.6  | 92.04  | 82.31 |
> | UniTE + SAFE (threshold = 0.3) | 77.21   |  76.6  | 91.89  | 81.90 |
> | UniTE + SAFE (geometric mean) | 78.27 |  77.6  | 92.27  | 82.71 |
>
> Thank you again for the helpful feedback, which strengthened our understanding of this component. We will incorporate these results and further comparative analysis into the final manuscript.

---

> > ### Author Response · Authors · 2025-12-03
> > **Response to Reviewer gPZv (W2)**
> >
> > **[W2: Additional response on the probability sharpening strategy]**
> >
> > We are pleased to report that the revised manuscript has now been updated to include the additional experimental results and analysis on probability sharpening, which were previously discussed in our response. Below, we summarize the newly incorporated content.
> >
> > In Sections 3.3 and 4.4 of the revised manuscript, we explore two different sharpening strategies and show that both strategies consistently improve performance. These results underscore the core takeaway of introducing sharpening: when ensembling models with heterogeneous tokenizers, the dispersed probability mass across multiple tokens could result in an overly smooth ensemble distribution, and incorporating sharpening effectively mitigates this issue.
> >
> > In Section 4.4, we include a detailed analysis of different sharpening strategies and the justification for our threshold choice. We observe that replacing the arithmetic mean with the geometric mean generally yields strong performance, which we attribute to the geometric mean’s ability to concentrate the dispersed probability mass on the token with the highest consensus across models. Furthermore, our probability sharpening strategy also consistently improves performance while preserving the widely adopted arithmetic mean ensemble form. Thanks to the general benefit of introducing sharpening, our framework allows each ensemble method to adopt the sharpening strategy most suitable for its characteristics. Finally, we have included the discussion regarding the threshold choice that was covered in our previous response.
> >
> > We sincerely thank the reviewer again for the thoughtful and constructive suggestion.

---

> ### Author Response · Authors · 2025-11-20
> **Response to Reviewer gPZv (2)**
>
> **[Q1: Does SAFE induce temperature-scaling-like effects that boost performance?]**
>
> We confirm that no strategies, such as temperature scaling or sampling, that could boost performance were used in our evaluation. All results were obtained strictly with greedy decoding. We also plan to release our code to facilitate further verification and ensure reproducibility.
>
> ---
> **[W3 + Q2: Why is vocabulary alignment expensive?]**
>
> We apologize for the insufficient explanation regarding vocabulary alignment. Our method improves the efficiency of ensemble methods in long-form generation through two factors:
>
> 1. restricting autoregressive generation to a single model, and
> 2. reducing the number of ensemble operations.
>
> Vocabulary alignment is related to the second factor, since performing ensemble operations requires aligning the next-token probability distributions defined over different vocabulary sets into a shared vocabulary space. This process introduces inefficiencey when generating long sequences, as it involves repeatedly mapping large vocabularies (e.g., ~150k tokens in Qwen) to the shared space.
>
> Regarding the writing, our intention was to convey that vocabulary alignment is a primary expense within the ensemble operation itself, not for the entire generation process. As the reviewer noted, the main source of efficiency comes from the first factor, namely restricting autoregressive generation to a single model. We recognize that our manuscript may have led readers to mistakenly believe that vocabulary alignment is a major expense for the entire generation process rather than only for the second factor, and we have revised the manuscript to avoid this confusion. Thank you for giving us the opportunity to clarify this point.
>
> ---
> **[W3 + Q3: Maximum memory usage of SAFE]**
>
> We measure the maximum memory usage of each method using `torch.cuda.max_memory_reserved()`. We report the maximum memory when generating 2048 tokens through ensembling Internlm3-8b-Instruct and Qwen2.5-7B-Instruct. Since each model is loaded on to a separate GPU, we measure the largest maximum memory usage among the GPUs.  All other experimental settings remain identical to those used in our main experiments. As shown in the below table, SAFE does not introduce significant additional memory overhead.
>
> | **Method** | **Maximum memory** |
> |-------------------|:--------------:|
> | GaC |17672 MB |
> | GaC + SAFE | 17636 MB |
> | UniTE | 17630 MB |
> | UniTE + SAFE | 17636 MB |
>
> ---
> **[W3 + Q4: Computational Overhead of Repeatedly Calling the Verifier Models’ Tokenizers During the Verify Step]**
>
> In the verification step, tokens that have already been accepted through ensembling are never modified, so their tokenizations can be cached. We only need to tokenize newly generated tokens from the drafter, which have not yet gone through verification. Under the same hardware setting used in our main experiments, when the drafter proposes 5 new tokens at a time, the tokenization cost per verifier is approximately *0.5ms*. When normalized per token, this corresponds to approximately *0.1ms/token*.
>
> ---
>
> Once again, we deeply appreciate your feedback and dedication. If you have any additional questions or comments, we would be happy to address them.

---

### Official Review · Reviewer_4xbi · 2025-10-29

**Soundness:** 3
**Presentation:** 2
**Contribution:** 3
**Rating:** 6
**Confidence:** 4

**Summary:**

This paper proposes a LLM Ensemble method, specifically categorized as an ensemble-during-inference method.
The proposed method uses a main model to generate fragment-level information, and then utilizes several other models as validators to check the "potential error-prone areas", and finally integrates the outputs of multiple models in these areas.

**Strengths:**

The proposed generate-verify-ensemble method, in terms of its overall approach and framework design (i.e., from a high-level perspective), is reasonable and intuitive.

**Weaknesses:**

**Weaknesses and Suggestions.**


1) The presentation of the method and experiments is not clear enough. This version of the paper requires significant revisions to make the presentation clearer and more accessible for readers.
Specifically: The presentation in the method section (Section 3) is not clear enough. Regarding the experiment section, for example: (i) In Section 4.2, when introducing the set of LLMs considered, mentioning the prior work proposed by Yao et al. is unnecessary and may cause confusion, as the LLM set in this paper differs from that in the prior work. (ii) The meanings of "GaC + SAFTE" and "UniTE + SAFE" are not clearly explained in the Section 4.2.

2) For the related work section, it is recommended to include an introduction to some classic works on speculative decoding, as the research in this paper is closely related to speculative decoding.

3) The main experimental results (Table 2) presented in this paper seem to show marginal performance improvement. It is recommended to include an overall average result in Table 2 to show the average performance improvement across all cases. Further experiment tuning and the design of new tricks could help improve the performance of the proposed method.

**Questions:**

Please refer to the above "Weaknesses and Suggestions" and the following:
1) The 5.72% performance improvement mentioned in the introduction—where is it reflected in the tables? The explanation of this in the experiment section seems unclear.
2) What aspect or meaning does the term "stable" in the paper's title primarily refer to? It would be helpful to explain and address this question from both the methodological and experimental result perspectives. It is recommended that the authors consider this issue in the final version of the paper.
3) If the authors intend to emphasize efficiency, it is recommended that they provide a comparison of the overall runtime of the proposed method and the baselines on these datasets.

---

> ### Author Response · Authors · 2025-11-20
> **Response to Reviewer 4xbi (1)**
>
> Dear Reviewer 4xbi,
>
> We sincerely appreciate the reviewer’s valuable feedback. We are confident that your comments have greatly helped us further improve the clarity and quality of our work. We are especially grateful for your positive assessment of the soundness of our approach. Below, we provide our responses to address the reviewer’s concerns.
>
> ---
> **[W1: Clarification of our method and experiments]**
>
> Thank you for your constructive feedback aimed at improving the quality of our paper. We have revised the manuscript to clarify the points you identified as potentially confusing to readers. We appreciate your suggestions, and we have incorporated the following major revisions to improve the presentation of our paper.
>
> **W1-1: Clarification of the method section**
>
> We have revised the manuscript to improve the clarity of the method section. In particular, the revised manuscript includes the following major updates:
>
> (1) In the Introduction (Section 1), we more clearly articulated our contributions and the advantages of SAFE.
>
> (2) In the Related Work section (Section 2.2), we added an explanation of speculative decoding to help readers who may not be familiar with this framework.
>
> (3) In the Verify step of our method (Section 3.2), we provided a more detailed explanation of how OOV-like tokens are identified, which is a key component of our approach.
>
> We are continuing to refine the manuscript to further improve clarity, and if any part remains unclear, please feel free to let us know at any time.
>
> **W1-2: Clarification of experimental settings**
>
> The reason we cited [1] in the section describing the models used in our experiments was solely to indicate that our model selection was guided by a similar motivation as in the prior work. However, we acknowledge that this reference may have caused confusion, since our actual experimental setup uses a different set of models to incorporate more recently released models. To avoid any misunderstanding, we have revised the manuscript to remove this unnecessary mention.
>
> [1] Yao, Yuxuan, et al. "Determine-then-ensemble: Necessity of top-k union for large language model ensembling.", ICLR 2025.
>
> **W1-3: The meanings of “GaC + SAFE” and “UniTE + SAFE”**
>
> Our work proposes a method for identifying appropriate token-level positions when generating long sequences, guided by the two key factors. At positions deemed appropriate for ensembling, existing ensemble methods can be applied directly, and this is precisely what “GaC + SAFE” and “UniTE + SAFE” represent. Our revised manuscript clarifies this point at the end of Section 4.1.

---

> ### Author Response · Authors · 2025-11-20
> **Response to Reviewer 4xbi (2)**
>
> **[W2: Adding Speculative Decoding to Related Work]**
>
> We agree that including speculative decoding in the related work section would help improve readers’ understanding, given the similarity between our paper and this line of research. Accordingly, we have added an explanation of speculative decoding in Section 2.2 of the revised manuscript.
>
> ---
> **[W3 + Q1: Average performance gain in Table 2.]**
>
> We additionally report the average performance gain in Table 2 in the revision. The gains shown in the table are computed relative to the best-performing individual model. However, our method achieves a larger improvement—approximately **5.72%**—when compared against the **baseline ensemble methods.**
>
> The 5.72% performance improvement represents the average performance gain of our approach over existing ensemble techniques in the two-model ensembling setting, based on the results in **Tables 2 and 6** of our revised manuscript (where Table 6 corresponds to Table 5 in the original submission). Since we evaluate three model pairs combined with two ensemble methods across five datasets, the reported average is computed over 30 total experiments (3 × 2 × 5). However, we acknowledge that this average is not directly observable from the tables. Therefore, to avoid confusion and improve clarity, we revised the statement from “outperforms existing ensembling methods by an average of 5.72% …” to “outperforms existing ensemble methods …”
>
> ---
> **[Q2: Meaning of “stable” in our title]**
>
> By “stable,” we intend to emphasize not only improved performance but also the ability to generate fluent text when ensembling models at the token-level. As shown in Figure 2 (Right), issues such as OOV-like tokens can lead to garbled outputs (e.g., SofÃ­aÂaÂÂÃ...­) during ensembling. In contrast, our method mitigates these OOV-like issues and enables stable text generation in the ensemble setting, which in turn also leads to improved performance across various benchmarks. We have revised the final part of the Introduction (line 137-140) to make this intention clearer. Thank you for giving us the opportunity to clarify this important point.
>
> ---
> **[Q3: Overall runtime of each dataset]**
>
> We did not report the overall runtime for each dataset because each method produces outputs of different lengths, which can substantially affect total runtime independent of their actual efficiency in ensembling. For this reason, we reported runtime as a function of output length, using all samples in each dataset. This follows the standard practice used in prior ensemble work [1,2,3].
>
> [1] Yao, Yuxuan, et al. "Determine-then-ensemble: Necessity of top-k union for large language model ensembling." ICLR 2025.
>
> [2] Yu, Yao-Ching, et al. "Breaking the ceiling of the llm community by treating token generation as a classification for ensembling." *EMNLP 2024*.
>
> [3] Huang, Yichong, et al. "Ensemble learning for heterogeneous large language models with deep parallel collaboration." *NeurIPS* 2024.
>
> ---
> Once again, we sincerely appreciate your time and effort. If you have any further questions or comments, we welcome any further discussion that may help strengthen the clarity and presentation of our work.

---

### Meta-Review · Area_Chair_avvN · 2025-12-17

**Summary:**

The paper proposes SAFE (Stable And Fast LLM Ensembling), a framework designed to improve the efficiency and performance of Large Language Model (LLM) ensembles during inference, specifically for long-form generation. The core innovation utilizes a speculative decoding approach where a "drafter" model generates a sequence, and "verifier" models determine if ensembling is necessary based on two criteria: (1) the absence of tokenization mismatches (avoiding "OOV-like" tokens) and (2) low consensus in next-token probability distributions. The method aims to mitigate the computational costs of vocabulary alignment and the degradation caused by incompatible tokenizers.

**Reviewer Concerns:**

The authors provided a comprehensive rebuttal that appears to have addressed the majority of the reviewers' technical and conceptual concerns through revised text and new experiments:

Conceptual Clarifications (Reviewers 4xbi, gPZv):
OOV-like vs. Non-Canonical Tokenization: Reviewer gPZv questioned the novelty of "OOV-like" tokens regarding existing literature on non-canonical tokenization. The authors clarified that "OOV-like" is strictly defined by the participating models' tokenizers during ensemble, whereas non-canonical covers all possible tokenizations. A discussion was added to Appendix H.
Meaning of "Stable": In response to Reviewer 4xbi, the authors clarified that "stable" refers to the method's ability to prevent garbled text generation caused by tokenizer mismatches, ensuring fluency.
Efficiency Claims: The authors clarified for Reviewer gPZv that the "cost of vocabulary alignment" refers specifically to the ensemble operation steps, not the entire generation pipeline, justifying why reducing these operations improves speed.

Experimental Validations (Reviewers gPZv, ux3H, vm3X):
Ablation Study: Provided for Reviewer ux3H, confirming that both OOV verification and Ensemble Distribution verification contribute independently to performance and efficiency.
Comparison to Baselines: ADDRESSED Reviewer vm3X’s concerns regarding CoS (Collaborative Decoding) and EVA. The authors explained that CoS requires identical tokenizers (making it inapplicable here) and empirically demonstrated that SAFE outperforms EVA’s confidence-based weighting.
Sharpening Strategies: In response to Reviewer gPZv, the authors tested alternative sharpening methods (e.g., geometric mean) and thresholds, demonstrating that the improvement is robust across different sharpening techniques.
Sampling Strategies: Addressed Reviewer vm3X’s concern about greedy decoding by demonstrating that SAFE works effectively with Top-K and temperature sampling.
Memory Overhead: Provided data for Reviewer gPZv showing SAFE introduces negligible memory overhead compared to baselines.


Reviewers' Concerns Not Addressed During the Rebuttal
Overall Runtime Table (Reviewer 4xbi): The reviewer requested a comparison of the overall runtime for datasets. The authors declined to provide a single aggregate number, arguing that runtime is a function of output length (which varies). They maintained their presentation of runtime as a function of output length. While methodologically sound, this technically leaves the specific request for an aggregate table unfulfilled.
Citations in Initial Submission: While rectified in the revision, the initial failure to cite "non-canonical tokenization" literature was a valid critique by Reviewer gPZv that required correction.

**Reviewer Scores:**

Current Scores: 6 (4xbi), 6 (gPZv), 8 (ux3H), 4 (vm3X).

Assessment: The paper is in a strong position for acceptance, potentially higher than the current average score suggests.

Reviewer ux3H (8) is positive but admits lower confidence.

Reviewers 4xbi and gPZv (6s) raised valid criticisms regarding clarity and definitions, which the authors appear to have resolved extensively in the revision (revised Method section, added definitions, new experiments). Their "marginal" acceptance might be conservative given the strength of the rebuttal data.

Reviewer vm3X (4) serves as the primary detractor, arguing that existing methods (CoS, Span-level) suffice. However, the authors provided a strong logical defense: CoS does not handle heterogeneous tokenizers, and span-level ensembling sacrifices token-level probability granularity. The authors backed this up with empirical data showing SAFE outperforms the specific baselines (EVA) suggested by the reviewer.

Recommendation: The rebuttal effectively dismantles the primary reasons for the negative score (4). Assuming the technical accuracy of the new experiments (Sampling, EVA comparison, Ablation), the paper demonstrates clear utility in the practical setting of ensembling open-source models with different tokenizers. The scores should likely trend upwards toward Accept.

---

### Decision · Program_Chairs · 2026-01-26

Accept (Poster)